# The potential of urban irrigation for counteracting carbon-climate feedback

Peiyuan Li [1,2], Zhi-Hua Wang [1] ✉ & Chenghao Wang [3,4]

Global climate changes, especially the rise of global mean temperature due to the increased carbon dioxide ($CO_2$) concentration, can, in turn, result in higher anthropogenic and biogenic greenhouse gas emissions. This potentially leads to a positive loop of climate–carbon feedback in the Earth's climate system, which calls for sustainable environmental strategies that can mitigate both heat and carbon emissions, such as urban greening. In this study, we investigate the impact of urban irrigation over green spaces on ambient temperatures and $CO_2$ exchange across major cities in the contiguous United States. Our modeling results indicate that the carbon release from urban ecosystem respiration is reduced by evaporative cooling in humid climate, but promoted in arid/semi-arid regions due to increased soil moisture. The irrigation-induced environmental co-benefit in heat and carbon mitigation is, in general, positively correlated with urban greening fraction and has the potential to help counteract climate–carbon feedback in the built environment.

Urban areas, covering only about 3% of global land surfaces, accommodate 56% of human population, consume over two-thirds of world's energy, and produce more than 70% of global carbon dioxide ($CO_2$) emissions[1,2]. Practically speaking, the sustainable future of human societies depends largely on urban sustainability. Today, the global urbanization, with concomitant burgeoning anthropogenic activities, has been the primary and the most irreversible driver to climate changes[3,4]. Critical challenges faced by cities in the context of global climate change include excessive heat stress, air pollution, public health risks, and degraded ecosystems, to name a few[5,6]. In particular, many urban environmental issues are strongly correlated with the warming at urban cores, a phenomenon known as the urban heat island (UHI) effect[7–9].

In addition, most anthropogenic heat emissions, such as those from vehicular and building operations, are also significant contributors to concentrated greenhouse gas (GHG) emissions, especially the anthropogenic $CO_2$ (An$CO_2$)[10,11], which is the dominant source of GHG forcing to climate changes[12]. The increasing $CO_2$ concentration produces rising global mean temperature, which in turn results in higher An$CO_2$ and biogenic $CO_2$ emissions by, e.g., more fossil fuel consumption for electricity generation during warm seasons[13] and

higher soil respiration rate[14], potentially leading to a positive loop of climate–carbon feedback in the Earth's climate system[15]. This highlights the critical importance for sustainable engineering solutions being capable of mitigating the compound environmental impact of coupled heat and carbon emissions[16], in order to effectively counteract the positive feedback loop.

In past decades, climate-resilient urban infrastructure designs, especially nature-based solutions, have been extensively studied aiming to mitigate the adverse environmental impacts concomitant with global urbanization[17,18]. In particular, urban greening, e.g., the use of lawns, trees, green roofs/walls, and urban irrigation, has been widely adopted in cities' climate action plans to mitigate heat stress as well as to reduce carbon emissions[19,20]. Despite the tremendous research effort, quantifying the comprehensive impact of urban greening on the Earth's climate, especially on heat and $CO_2$ emissions, remains challenging to researchers given the large spatiotemporal uncertainties of biogenic $CO_2$ exchange (i.e., $CO_2$ uptake and release via photosynthesis and respiration processes, respectively)[21–23] and the very limited urban observations on biogenic sectors at global scale[24]. Recent reviews also found significant gaps in implementing nature-based solutions, such as the uncertainties on the effectiveness of the

[1]School of Sustainable Engineering and the Built Environment, Arizona State University, Tempe, AZ, USA. [2]Discovery Partners Institute, University of Illinois System, Chicago, IL, USA. [3]School of Meteorology, University of Oklahoma, Norman, OK, USA. [4]Department of Geography and Environmental Sustainability, University of Oklahoma, Norman, OK, USA. ✉e-mail: zhwang@asu.edu

solution[25] and lack of public awareness and trustworthy information[26], which cause operational barriers for stakeholders. Overcoming these challenges requires a comprehensive understanding of the interplays between urban vegetation and its surroundings. This further leads to the necessity of investigating the compound environmental feedback of urban green spaces[16].

$CO_2$ exchange of urban greenery is passively influenced by warming from UHI[27] and $CO_2$ fertilization effect[28] from anthropogenic emissions. These factors, as concluded in prior studies, generally enhance the growth of urban plants, and thus improve carbon sequestration[29,30]. In addition, active landscaping management on urban greenery, such as irrigation, affects $CO_2$ exchange as well. Observational-based studies showed unintended $CO_2$ release due to irrigation, especially over-managed green spaces such as lawns, gardens, and golf courses[31–33], whereas a recent modeling study suggested the potential environmental co-benefits in heat mitigation and carbon reduction[34]. These contrasting findings imply the critical role of irrigation in controlling the direction of $CO_2$ fluxes and its significant spatial variability. Despite these insights regarding the irrigation-induced impact on $CO_2$ fluxes, the aforementioned studies were considered inadequate compared with the extensively studied irrigation-induced cooling[35,36]. Moreover, urban irrigation accounts for half of the residential water use[37], and this number can escalate to 80% in semi-arid/arid regions like some cities in California[38]. Many cities have demonstrated their determinations in reducing outdoor water use via water conservation programs[39]. However, it is equally important to not neglect other environmental benefits from irrigation beyond cooling. As of now, the underlying mechanisms that govern the water–heat–carbon dynamics, along with the drastic spatial variations of the irrigation-induced impacts, remain largely obscure. This knowledge gap hinders the evidence-based decision-making process, which motivates further investigations on the impact of urban irrigation on a larger spatial scale.

In this study, we develop a modeling framework by resolving the dynamics of biogenic $CO_2$ exchange in the built environment based on the Weather Research and Forecasting (WRF) model[40,41] and an advanced urban canopy scheme, viz. the Arizona Single-Layer Urban canopy Model (ASLUM)[42]. This new modeling tool is then applied to unravel the complex interactions between heat and carbon dynamics in the urban environment. In particular, we aim to investigate the impact of urban irrigation, as one of the most popular and widely studied climate mitigation strategies, on modifying urban climate as well as the response of urban ecosystem exchange in different climate regions in the U.S. under real meteorological conditions. In addition, we unravel the mechanistic pathways that control the biogenic fluxes over urban green spaces, which enables urban researchers and policy makers to identify the potential tradeoffs in the compound heat–carbon exchange processes, and to achieve a net environmental co-benefit towards more sustainable climate change mitigation.

## Results and discussion
### The cooling effect of urban irrigation
The impact of urban irrigation is quantified from the difference between the irrigation case and the baseline case. In our model, urban irrigation is conducted during 21:00–22:00 local time every day and will cease once the soil water content reaches a prescribed threshold (see Method). Irrigation-induced cooling effect is manifest with a modest spatial variation. Figure 1 shows the change of daily mean 2-meter air temperature ($dT_{2m}$) over the contiguous U.S. (CONUS) and 12 major metropolitan regions. On average, urban irrigation cools the cities and their surrounding area by 0.26 °C, with the most pronounced cooling effect in Salt Lake City, UT (−0.59 °C), followed by Dallas-Fort Worth, TX (−0.50 °C) and Phoenix, AZ (−0.48 °C). During the simulation period, there are two regional heatwaves on record in the US, namely the 2013 Southwest heatwave (June 29th to July 2nd)

and 2015 Northwest heatwave (June 26th to June 28th). The cooling effect from irrigation is more significant during heatwaves than normal summer days (CONUS: −0.32 °C; Salt Lake City, UT: −0.74 °C; Dallas-Fort Worth, TX: −0.6 °C; Phoenix, AZ: −0.73 °C). In contrast, surface soil cooling by irrigation ($dT_{soil}$) is more substantial compared with the cooling of air (Supplementary Fig. 1 cf. Fig. 1). The average soil temperature drops 1.84 °C over urban areas and the reduction varies across the CONUS. The most significant soil cooling happens in Salt Lake City, UT (−4.66 °C), followed by Los Angeles, CA (−4.08 °C) and Phoenix, AZ (−3.97 °C).

It is noteworthy that the relatively conservative irrigation applied in this study (see Method) will not generate surface runoff or excessive soil water. This treatment mimics the operation of on-demand irrigation system for water conservation. Air cooling in this baseline scenario is subtle due to the positive correlations between the cooling magnitude and irrigation amount[36,43,44]. The spatial variations of air cooling are rather limited as well. The surface and soil cooling, on the other hand, is more pronounced and comparable to the previous study[43]. We observe that cities in arid climate regions generally have more noticeable soil cooling, owing to the high atmospheric demand in the arid environment[36].

The levels of air and soil cooling collectively affect carbon balance over urban vegetation through intricate soil–plant–atmosphere interactions (Fig. 2a). Qualitatively, photosynthesis process in plant leaves is directly affected by the change of air temperature (Fig. 2b); while the change of soil temperature mainly influences the respiration processes occurring in plant roots and soil (Fig. 2c). Under normal climate without extreme heat or drought conditions, irrigation-induced cooling will likely cause the photosynthesis and respiration processes to depart from their optimum reaction conditions, leading to the decreases of the urban gross primary productivity ($GPP_u$), i.e., the sum of $CO_2$ uptake via photosynthesis, and the urban ecosystem respiration ($R_u$), i.e., the sum of $CO_2$ release via respiration processes. Meanwhile, the different degrees of cooling in the air and soil will lead to more drastic changes of $R_u$ than $GPP_u$. On the contrary, apart from cooling, irrigation enriches soil moisture and promotes the reaction rates of photosynthesis and respiration. Interestingly, the cooling and moisturizing effects influence $CO_2$ exchange in opposite ways, governed by a pair of adverse mechanisms. Urban net ecosystem exchange ($NEE_u$) is then highly dependent on the various pathway and synthesis of temperature–moisture–carbon interactions (Fig. 2d). The actual outcome is a complex function of prevailing anthropogenic, geographic, and climatic conditions in the built environment with strong locality.

### The impact of urban irrigation on $CO_2$ exchange
Figure 3 shows the irrigation-induced change of net ecosystem exchange ($dNEE_u$) over CONUS and 12 major urbanized regions. Surprisingly, additional 0.22 g m$^{-2}$ d$^{-1}$ $CO_2$ is released to the atmosphere from irrigated urban greenery over CONUS, leading to an overall negative impact. Nevertheless, the spatial distribution of $dNEE_u$ exhibits distinctive patterns over the east–west extent. Most cities in western U.S. show increases in $dNEE_u$, while eastern cities generally show decreases with a few exceptions such as Boston, MA, Charlotte, NC, and Pittsburgh, PA.

When examining the two components in $dNEE_u$, i.e., the changes of $GPP_u$ ($dGPP_u$, Fig. 4) and $R_u$ ($dR_u$, Fig. 5), we find that the daily mean $GPP_u$ and $R_u$ over CONUS increase by 0.19 gCO$_2$ m$^{-2}$ d$^{-1}$ and 0.41 gCO$_2$ m$^{-2}$ d$^{-1}$, respectively, but with different spatial patterns. The change of $GPP_u$ is evenly distributed across the CONUS with a relatively small variation from 0.05 gCO$_2$ m$^{-2}$ d$^{-1}$ (Phoenix, AZ) to 0.52 gCO$_2$ m$^{-2}$ d$^{-1}$ (Seattle, WA) among the 12 major urbanized regions (Fig. 4). We also notice that $dGPP_u$ is positively correlated with vegetation fractions. Cities with large increase of $GPP_u$, such as Seattle, WA (0.52 gCO$_2$ m$^{-2}$ d$^{-1}$), Houston, TX (0.41 gCO$_2$ m$^{-2}$ d$^{-1}$), and Portland, OR

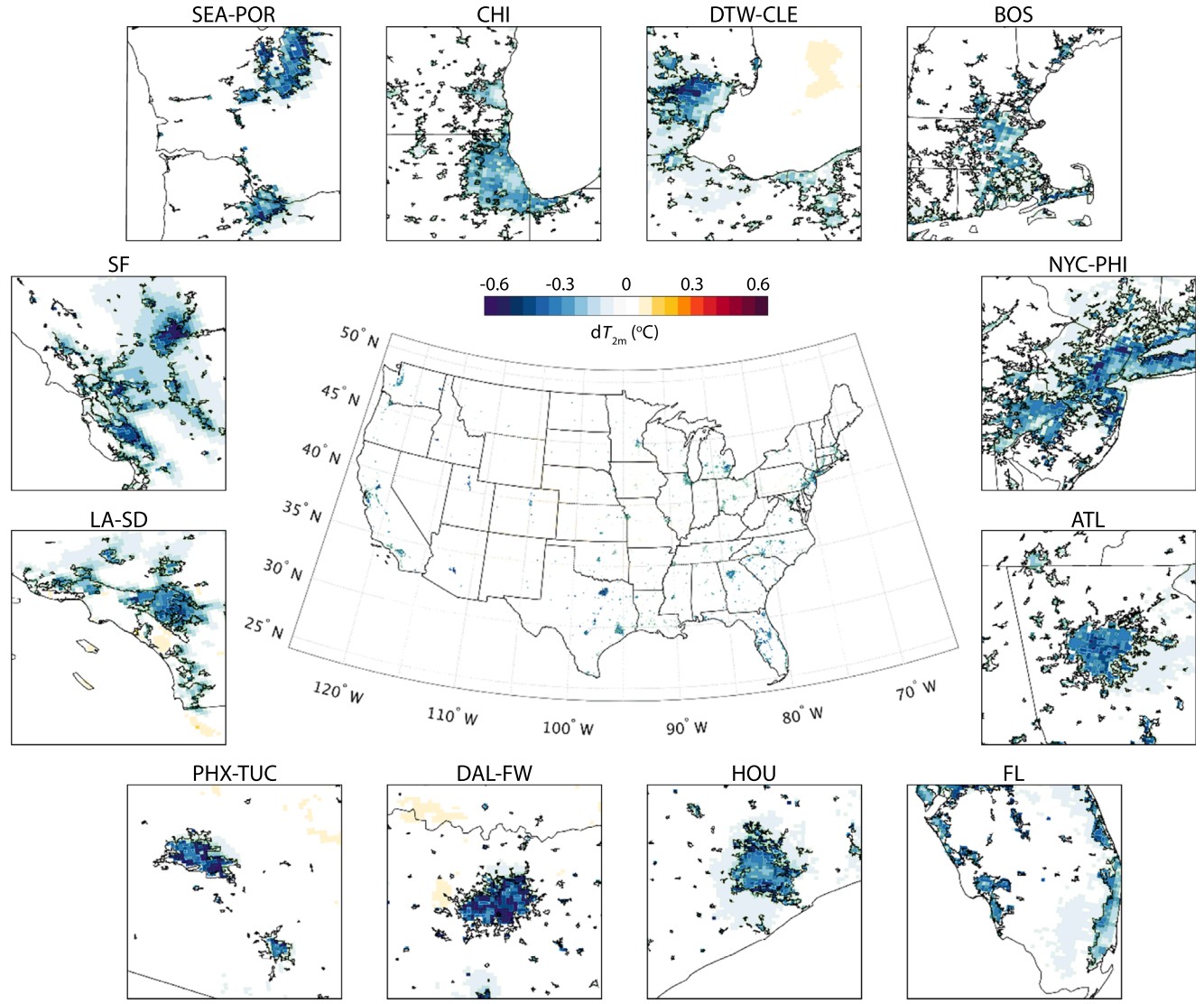

**Fig. 1 | Changes of simulated daily mean 2-meter temperature (d$T_{2m}$) after applying urban irrigation.** Subplots around the CONUS map show the details over 12 metropolitan regions. SEA-POR: Seattle, WA and Portland, OR; CHI: Chicago, IL; DT-CLE: Detroit, MI and Cleveland, OH; BOS: Boston, MA; NYC-PHI: New York, NY and Philadelphia, PA; ATL: Atlanta, GA; FL: Cities around the coast of Florida; HOU: Houston, TX; DAL-FW: Dallas and Fort Worth, TX; PHX-TUC, Phoenix and Tucson, AZ; LA-SD: Los Angeles and San Diego, CA; SF: San Francisco, CA.

(0.33 gCO$_2$ m$^{-2}$ d$^{-1}$), have vegetation coverage over 55%. This correlation is also reflected within the metropolitan regions, represented by the declining gradients of dGPP$_u$ in Los Angeles, CA, Phoenix, AZ, Chicago, IL, and Detroit, MI from their urban cores to the city outskirts.

The spatial variation of the change in R$_u$ is much greater than that in GPP$_u$ across the CONUS (Fig. 5). We find west coast cities have the largest R$_u$ increase after irrigation, such as Los Angeles, CA (2.33 gCO$_2$ m$^{-2}$ d$^{-1}$), San Jose, CA (2.11 gCO$_2$ m$^{-2}$ d$^{-1}$), and Seattle, WA (1.23 gCO$_2$ m$^{-2}$ d$^{-1}$). The increase of R$_u$ is less apparent in southwest and south cities, such as Houston, TX (1.03 gCO$_2$ m$^{-2}$ d$^{-1}$), Salt Lake City, UT (0.86 gCO$_2$ m$^{-2}$ d$^{-1}$), and Phoenix (0.67 gCO$_2$ m$^{-2}$ d$^{-1}$). In contrast, some eastern cities show decreases of R$_u$, such as Philadelphia, PA (−0.44 gCO$_2$ m$^{-2}$ d$^{-1}$), New York, NY (−0.34 gCO$_2$ m$^{-2}$ d$^{-1}$), and Chicago, IL (−0.26 gCO$_2$ m$^{-2}$ d$^{-1}$). Although the magnitude of these decreases is not as notable as that of the increases, the green spaces in these cities release less CO$_2$ after irrigation, making positive contributions to carbon reduction. It is worth mentioning that the increased R$_u$ in Boston, MA (0.53 gCO$_2$ m$^{-2}$ d$^{-1}$) and Phoenix, AZ (0.67 gCO$_2$ m$^{-2}$ d$^{-1}$) agree with previous literature[31,33]. The greater variation of dR$_u$, as well as its larger mean value, contributes to the notable change of NEE$_u$

over the CONUS. Therefore, we see similar spatial patterns between dR$_u$ and dNEE$_u$ (Figs. 3 and 5).

Due to the major role of R$_u$ in the overall carbon exchange, we further investigate the governing processes of dR$_u$ in different cities. For example, Phoenix, AZ experiences significant soil cooling in our experiment (Supplementary Fig. 1), which theoretically suppresses respiration rate. But the additional soil moisture from irrigation promotes the biochemical reactions and offset carbon reduction from cooling. For clarity, we treat R$_u$ as a partial function of soil temperature and soil water content (Fig. 2e). We then define the terms $\partial R_u/\partial SWC$ and $\partial R_u/\partial T_{soil}$ to represent the change of R$_u$ induced by the change of soil water content, and the change of R$_u$ by the change of soil temperature, respectively. By definition, both terms are positive throughout the range of the variables in this discussion. If $\partial R_u/\partial SWC > \partial R_u/\partial T_{soil}$, R$_u$ will be mainly influenced by soil water content and will tend to increase after irrigation (soil water dominant process) (Fig. 2c). Otherwise, R$_u$ will be determined by soil cooling and will decrease after irrigation (temperature dominant process). Among 20 major US cities, eleven have increased R$_u$ (dR$_u$ > 0, red circles in Fig. 2e) after irrigation, while the rest nine have decreased R$_u$ (dR$_u$ < 0, yellow

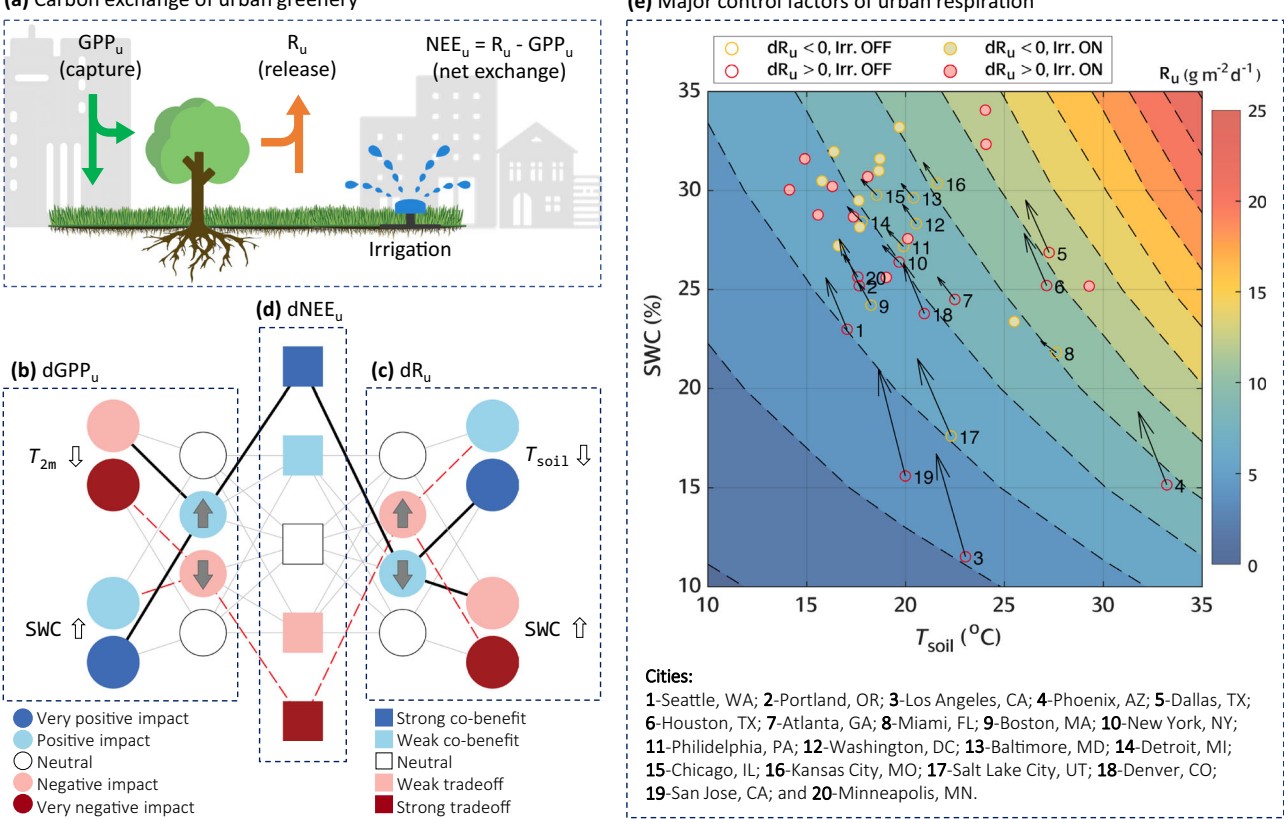

**Fig. 2 | Governing mechanisms on carbon exchange of urban greenery. a** A diagram showing carbon exchange of plants in the built environment with UHI, higher background $CO_2$ concentration, and management (irrigation). **b** Irrigation-induced change of urban gross primary productivity (d$GPP_u$), led by decrease of air temperature and increase of soil water content. **c** Irrigation-induced change of urban ecosystem respiration (d$R_u$), led by decrease of soil temperature and increase of soil water content. **d** Irrigation-induced change of urban net ecosystem exchange (d$NEE_u$), resulting from the combinations of d$GPP_u$ and d$R_u$. **e** Urban ecosystem respiration as a function of soil temperature and soil water content. The light gray lines in **b**–**d** show all possible combinations lead to various types of outcomes. The black solid lines indicate pathway to the strong co-benefit effect. The red dashed lines indicate the pathway to the strong tradeoff effect. Circles in (**e**) indicate the average $R_u$ before (hollow) and after (solid) irrigation. Arrows indicate the direction of change.

circles in Fig. 2e). Phoenix, for instance, experiences a soil water dominant process ($\partial R_u/\partial SWC > \partial R_u/\partial T_{soil}$), meaning that the $R_u$ improved by soil water outpaces the $R_u$ reduced by cooling. However, other cities, such as Chicago, IL, New York, NY, Philadelphia, PA, and Baltimore, MD, have the temperature dominant process ($\partial R_u/\partial SWC < \partial R_u/\partial T_{soil}$), leading to the overall decrease of $R_u$ (Fig. 2e). For cities like Atlanta, GA and Miami, FL, $\partial R_u/\partial SWC$, and $\partial R_u/\partial T_{soil}$ are roughly the same, thus the change of $R_u$ is minor (Fig. 5).

**Environmental co-benefit or tradeoff**
We find that urban irrigation affects $NEE_u$ primarily via its impact on respiration rate, which is governed by either soil water or temperature dominant process. With the ubiquitous cooling, cities with reduced $NEE_u$ experience environmental co-benefit from irrigation. However, the general trend in CONUS indicates that greater cooling happens with the cost of additional $CO_2$ release as an environmental tradeoff. Figure 6a, b shows dependence of the change in $NEE_u$ on air (d$T_{2m}$) and soil cooling (d$T_{soil}$) over US cities, respectively. The overall relationships exhibit tradeoff effects. With the prescribed irrigation, the regression over the 20 major cities indicates additional 4.15 $gCO_2 \, m^{-2} \, d^{-1}$ will be released per 1 °C of air cooling (dark blue dashed line in Fig. 6a). This number drops to 0.71 $gCO_2 \, m^{-2} \, d^{-1} \, °C^{-1}$ for soil temperature because of the higher efficiency of irrigation on soil cooling (black dashed line in Fig. 6b). Both tradeoff relations are statistically significant ($p < 0.005$, $R^2 > 0.40$). On the contrary, the dependence of d$NEE_u$ on d$T_{2m}$ over-all urban areas in CONUS is less significant with $R^2 = 0.02$ (red dashed line in Fig. 6a). We observe many

points densely clustered in the co-benefit quadrant between d$T_{2m} > -0.4$ °C and $-2 \, gCO_2 \, m^{-2} \, d^{-1} <$ d$NEE_u < 0$ in Fig. 6a. These points also correspond to the clusters in Fig. 6b between $-4$ °C $<$ d$T_{soil} < -2$ °C and $-2 \, gCO_2 \, m^{-2} \, d^{-1} <$ d$NEE_u < 0$. Despite subtle cooling and carbon reduction, these regions exhibit environmental co-benefit. This phenomenon is rarely observed when soil temperature drop is beyond $-4$ °C.

Meanwhile, we find significant positive relation between d$NEE_u$ and the change of soil water content (dSWC) (Fig. 6c), indicating the positive contribution from the rich soil water to additional $CO_2$ release from urban greenery. $NEE_u$ can be sensitive to the moisturizing effect as 1.46 $gCO_2 \, m^{-2} \, d^{-1}$ will be released per 0.1 unit increase of SWC (dark dashed line in Fig. 6c). For example, although Chicago, IL and Boston, MA have similar air and soil cooling, Boston, MA has a relatively higher percentage increase of SWC after irrigation (Fig. 2e). Therefore, irrigation causes a co-benefit in Chicago, IL but a tradeoff in Boston, MA.

Moreover, we classify the cities into 5 categories according to the changes of $NEE_u$ (Fig. 2d). Cities like Philadelphia, PA, and Baltimore, MD exhibit strong co-benefit effect, resulting from the simultaneous increase of $GPP_u$ and decrease of $R_u$ (black solid lines in Fig. 2d); while Los Angeles, CA and San Jose, CA release the most $CO_2$ after irrigation, exhibiting strong tradeoff (red dash lines in Fig. 2d). Cities with moderate changes of $NEE_u$ that result from various mechanistic pathways show weak co-benefit or weak tradeoff depending on the contributions from the increase of $GPP_u$ and the decrease of $R_u$. For example, both Atlanta, GA, and Chicago, IL show weak co-benefit, but d$GPP_u$ contributes 79% of the d$NEE_u$ in Atlanta, meaning the cooling and

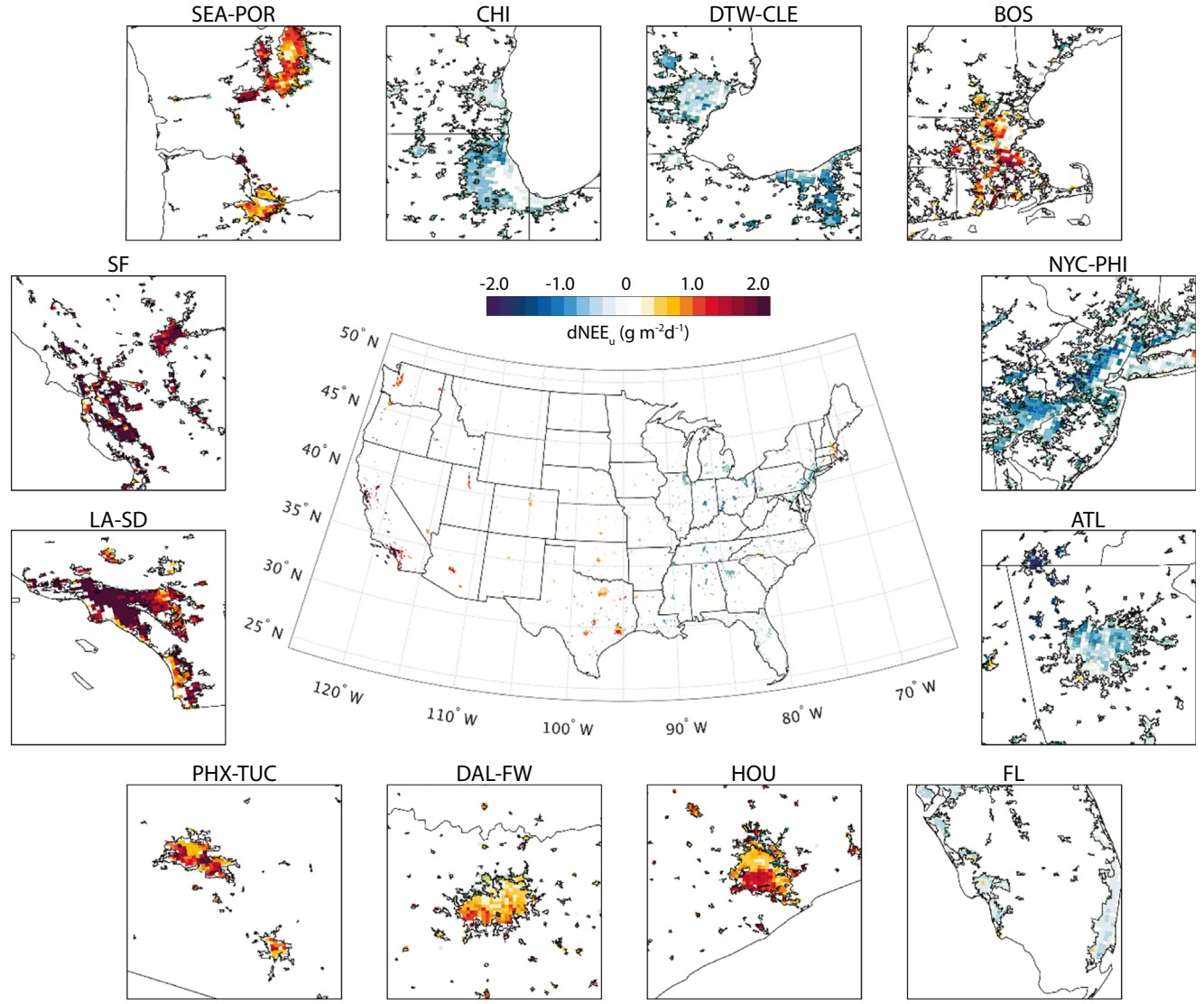

**Fig. 3 | Change of daily mean urban net ecosystem exchange (dNEE$_u$) after irrigation.** Subplots around the CONUS map show the dNEE$_u$ over 12 major metropolitan regions with the same acronyms as Fig. 1. The color bar is set to show positive environmental impacts in cool colors (green and blue), while negative impacts are shown in warm colors (yellow and red).

moisturizing effects nearly offset each other when affecting $R_u$. The contribution from dGPP$_u$ drops to 29% in Chicago, indicating the offset mainly happens in GPP$_u$. On the contrary, weak tradeoff occurs when a minor increase of GPP$_u$ is offset by a significant increase of $R_u$, such as Los Angeles, CA and Phoenix, AZ. The other pathway leading to tradeoff effect requires significant decrease of GPP$_u$, which is not observed in our simulation.

It is noteworthy that the occurrence of environmental co-benefit is conditioned on the cooling magnitude, irrigation amount, pre-existing soil water, etc., represented by various mechanistic pathways and the large spatial variabilities in the outcomes over the CONUS. To avoid unintended carbon release and achieve strong co-benefit effect, one needs to precisely tune the irrigation according to the local climatic and land use characteristics to increase the carbon sequestration from GPP$_u$ while suppressing $R_u$ (black solid lines in Fig. 2d). Specifically, irrigation should promote GPP$_u$ via moisturizing but also avoid the reduction on GPP$_u$ due to air cooling; on the other hand, it is also possible to control $R_u$ by improving soil cooling efficiency while keeping respiration change as a "temperature dominant process". Nonetheless, the climate of regions with high cooling efficiency is usually warm and arid. Their carbon environment tends to be more sensitive to the change of soil water. In this case, additional urban greenery strategies, such as shading trees and lawn expansion, will cool the environment and add biomass for photosynthesis, thus are more efficient in achieving the co-benefit[34]. For more humid regions that suffer less from water stress, urban irrigation needs to be rigorously regulated to avoid excessive soil water, especially during the wet season. It is recommended to equip the irrigation system with soil moisture sensors in residential yards, urban parks, and other maintained landscapes to optimize water use for conservation and carbon reduction purposes.

## Implications for carbon reduction

The result shows that irrigation reduces the net sequestration rate from urban vegetation by 0.22 gCO$_2$ m$^{-2}$ d$^{-1}$, primarily due to the unintended increase of soil respiration. This irrigation-induced change in CO$_2$ flux is very significant, in comparison to the latest estimate of the posterior 5-year annual mean global land atmospheric CO$_2$ growth rate of 0.41 gCO$_2$ m$^{-2}$ d$^{-1}$ (5.35 PgC yr$^{-1}$ over global land area)[45]. In particular, the change is more notable when comparing it with the ongoing carbon reduction effort in the anthropogenic sector. For example, replacing a gasoline vehicle by a battery electric vehicle (BEV) can save 4.4 tons CO$_2$

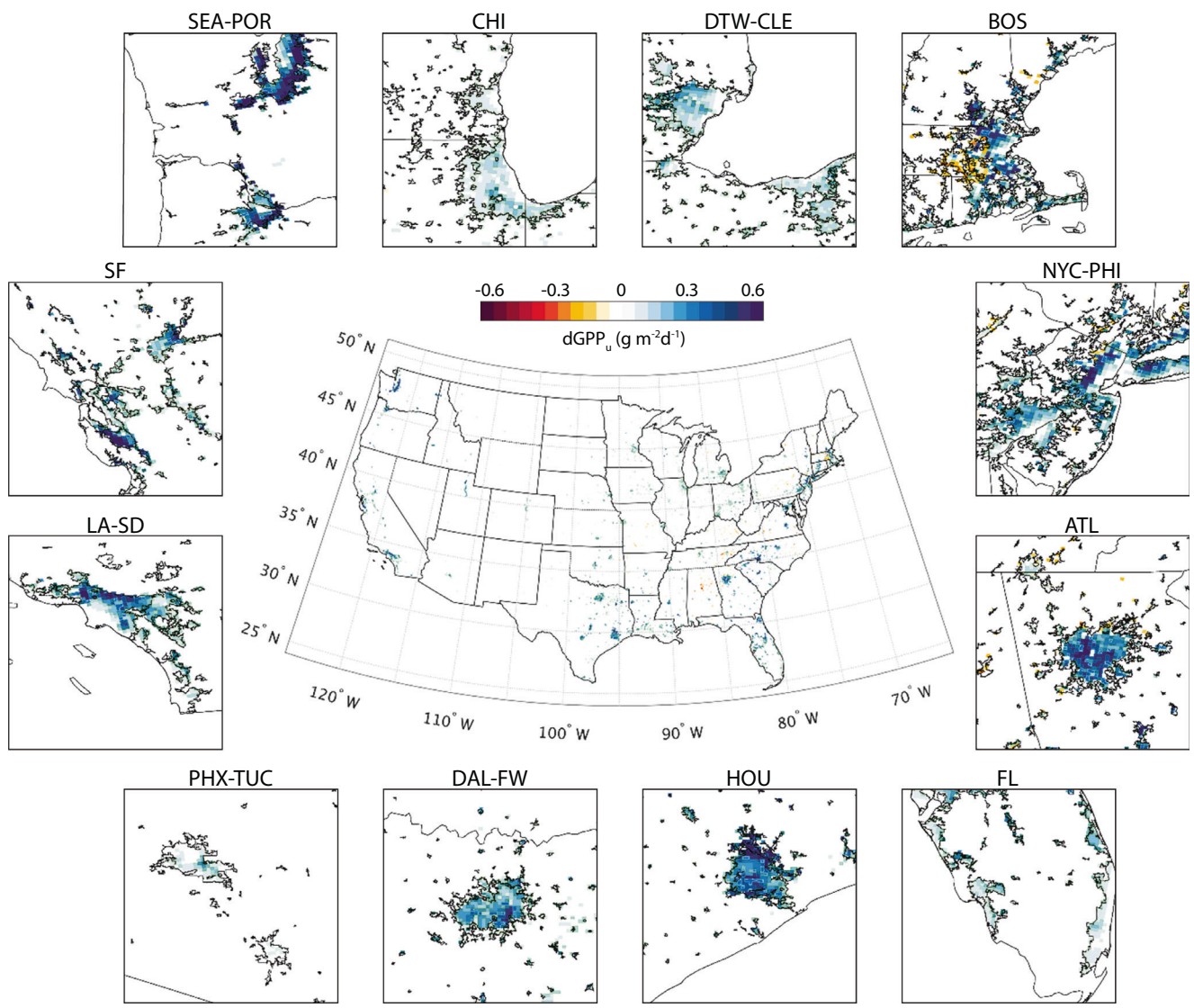

**Fig. 4 | Change of daily mean urban gross primary productivity (dGPP_u) after irrigation.** Subplots around the CONUS map show the dGPP_u over 12 major metropolitan regions with the same acronyms as Fig. 1. The direction of color bar is adjusted from Fig. 3 to consistently show positive environmental impacts in cool colors (green and blue), while negative impacts are shown in warm colors (yellow and red).

per year[46] due to the reduced on-road emission (i.e., without accounting the carbon emissions embedded in EV manufacturing and charging infrastructures). In this case, assuming 20% of the total urban land in the US implements irrigation as modeled, the unintended release will accumulate to $8.9 \times 10^3$ tons $CO_2$ per day or $1.1 \times 10^6$ tons $CO_2$ per summer, which offsets the effort of replacing a quarter million gasoline cars with BEVs (see Method). Conversely, cities with increased sequestration rates such as Chicago, IL, and Atlanta, GA, can benefit significantly from urban irrigation, which are equivalent to the carbon reductions from adopting ~8800 and 16,100 BEVs, respectively. Note that the annual BEV sales in US are 0.6 million and 0.8 million units in 2021 and 2022. Despite the significant effort to incentivize EV adoption at both the city and federal levels, there is still no consensus on the crucial role of urban green spaces in carbon budgeting.

Urban greenery and its irrigation amount a popular form of nature-based solutions for promoting environmental quality, which is also envisioned to work synergistically with local and regional environmental determinants to achieve optimal efficacy. However, we find that neither the potential for carbon reduction is fully explored, nor are the unintended consequences well recognized. The results of this study call for further effort to optimize irrigation schemes to

counteract the climate–carbon feedback by maximizing the environmental co-benefit or reducing the tradeoff of heat–carbon mitigation. In addition to the ambient temperatures and biogenic carbon exchange considered in this study, a holistic measure of urban greening and irrigation should include other relevant environmental and sustainable indicators, such as the efficiency of water and energy uses, air quality, human thermal comfort and health risks, and ecosystem services[16]. Yet, disentangling the coupled dynamics of heat, moisture, and carbon exchanges in the built environment imposes significant challenges upon the prevailing modeling and operational frameworks that are often exclusively designed to evaluate a singular strategy (heat mitigation in particular) at a time. It is therefore imperative to improve the capabilities of the physically-based urban models for a more comprehensive representation of the built environment, for the holistic evaluation of compound environmental impacts, especially those attributable to the climate–carbon feedback and responsible for anthropogenically-induced climate changes.

## Limitations and future work
Findings from the proposed modeling framework in this study can be informative for the evidence-based decision-making towards

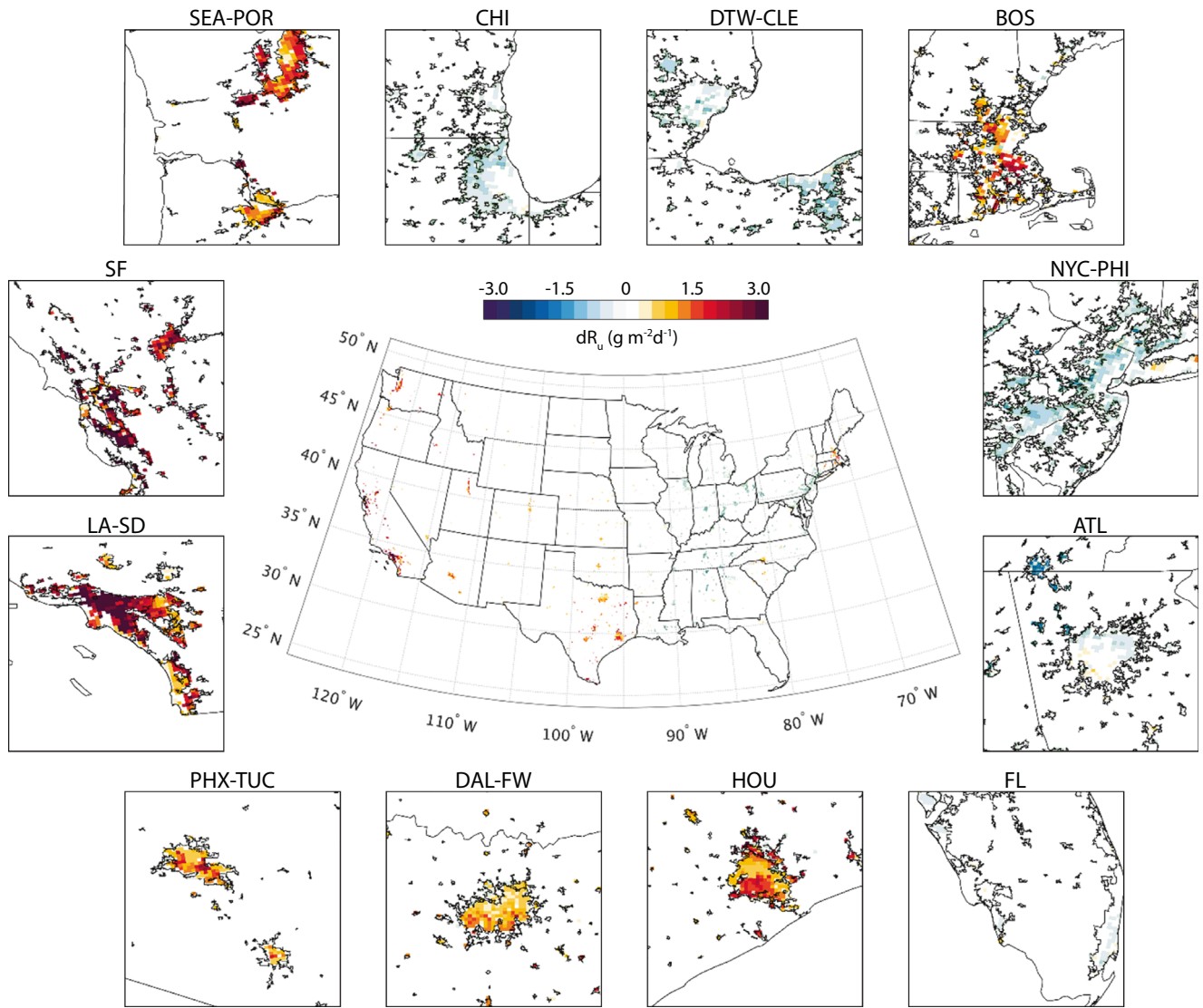

**Fig. 5 | Change of daily mean urban ecosystem respiration (dR$_u$) after irrigation.** Subplots around the CONUS map show the dR$_u$ over 12 major metropolitan regions with the same acronyms as Fig. 1. The color bar is set to be the same as Fig. 3 to consistently show positive environmental impacts in cool colors (green and blue), while negative impacts are shown in warm colors (yellow and red).

sustainable city development. Nevertheless, we reckon a few caveats of this study, primarily caused by the constraints of data availability and computational resources. More specifically, the lack of observation-based datasets on urban ecosystem services imposes great challenges to estimate long-term urban biogenic $CO_2$ exchange at a large spatial scale and hinders the validation of such numerical models. The scarcity of urban data on biogenic sectors is partially due to the difficulties of measurement in the highly heterogeneous urban environment. It is more likely, however, led by the lack of consensus on the significance role of urban vegetation and the impact of anthropogenic forcings on green spaces. The dearth of urban data further hinders the modeling endeavor to accurately quantify the biogenic carbon exchange in cities, making it even more difficult to simulate the plant response under the controls of the built environment. Our modeling framework, albeit at its infancy, contributes to filling this research gap, which can further aid the overall evaluation of the efficacy of nature-based solutions and guide the execution of city-level climate action plans. Based on the key findings of this study, with caveat, we recommend the urban research community to devote more effort on urban observation and data synthesis over biogenic sectors.

On the other hand, urban green spaces can be highly fragmented, which need to be simulated at a much finer spatial scale. The realistic parameterization and assessment of nature-based solutions inevitably involves high dimensionality of physical and modeling spaces due to the complexity of dynamic processes and the spatiotemporal heterogeneity of geographic and climate conditions. For a process-based model, it is of vital importance to balance the model complexity and feasibility, sometimes with the sacrifice of neglecting secondary processes such as plant fertilization or excluding the variances led by different plant species. The current simulation may not be able to fully capture these detailed heterogeneity and processes due to the lack of information to fine-tune the model parameters for different cities across the US. This will inevitably lead to uncertainties of the modeling results. However, in the face of a changing climate, more complex urban system dynamics as well as the optimization of its environmental solutions must be considered; therefore, it requires a new system-based, rather than process-based, paradigm in future generation of urban climate models. The development of such a paradigm should include, for example, complex network analysis (e.g., to identify the clustering or core–periphery structure of CONUS urban networks[47]), physical emergence (e.g., to identify abrupt and

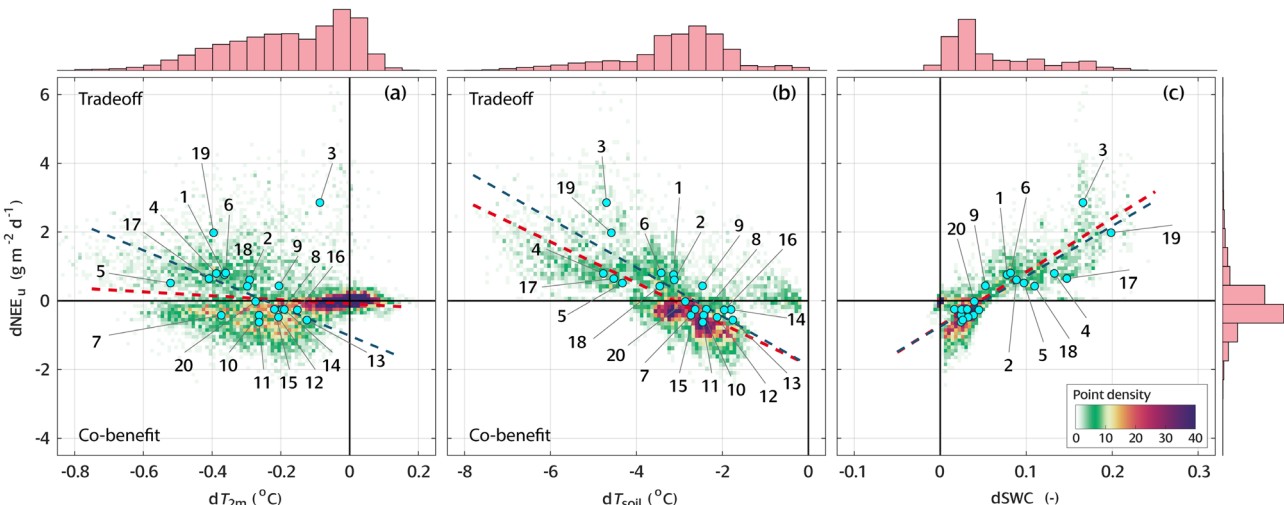

**Fig. 6 | Dependences of urban net ecosystem exchange (dNEE$_u$) on temperatures and soil water content in CONUS. a** Dependence of dNEE$_u$ on the change of 2-meter air temperature (d$T_{2m}$). **b** On the change of soil temperature (d$T_{soil}$). **c** On the change of soil water content (dSWC). The background scatters in (**a**–**c**) show relation in all urban cells from the model. The cyan solid circles show the average over 20 major cities in CONUS (see Fig. 2e for city number). The red dashed lines show the linear regression of the background scatters: **a** dNEE$_u$ = −0.57 d$T_{2m}$ −0.08, $R^2$ = 0.02, $p$ < 0.0001; **b** dNEE$_u$ = −0.60 d$T_{soil}$ −1.87, $R^2$ = 0.48, $p$ < 0.0001;

**c** dNEE$_u$ = 15.59 dSWC −0.72, $R^2$ = 0.68, $p$ < 0.0001. The dark blue dashed lines show the linear regression over the 20 cities: **a** dNEE$_u$ = −4.15 d$T_{2m}$ −1.02, $R^2$ = 0.41, $p$ = 0.0034; **b** dNEE$_u$ = −0.71 d$T_{soil}$ −1.87, $R^2$ = 0.65, $p$ < 0.0001; **c** dNEE$_u$ = 14.62 dSWC −0.75, $R^2$ = 0.79, $p$ < 0.0001. In **a** City 3-Los Angeles, CA is excluded as an outlier. All other regressions use all data. $p$-values indicate the significance of the regression coefficient. The histograms above and on the right show the distributions of d$T_{2m}$, d$T_{soil}$, dSWC, and dNEE$_u$, respectively.

potentially catastrophic transitions in nonlinear the climate system[48]), advanced machine learning techniques (e.g., for optimizing multiple environmental measures simultaneously[49]), to name a few. These system-based approaches are largely data-driven, and will benefit from the growing availability of datasets from measurements and re-analysis, such as the street-level monitoring networks, aerial imaging (such as LiDAR), high-resolution remote sensing, campaigns, and surveys.

## Methods

### Urban biogenic CO$_2$ exchange modeling

We adopt <u>A</u>rizona <u>S</u>ingle-<u>L</u>ayer <u>U</u>rban canopy <u>M</u>odel (ASLUM) version 4.0 in this study as the land surface scheme for the estimations of microclimate conditions and biogenic CO$_2$ exchange in the built environment[42]. ASLUM integrates the urban thermal and hydrological processes using a single-layer urban canopy scheme[50], and is under continuous development in the past decade[51–54]. In ASLUM, the urban area is represented as a two-dimensional (2D) street canyon, consisting of two arrays of buildings separated by a road, with infinite longitudinal dimensions. The geometric dimensions are configured by canyon width ($w$), building height ($h$), and building roof width ($r$). Combined with the land use portfolio, these parameters determine the redistribution of surface available energy and thus the in-canyon microclimate. ASLUM version 4 is capable of resolving a holistic set of urban CO$_2$ uptake and emission arising from various sources, including human, building, and vehicular CO$_2$ emissions, plant biogenic CO$_2$ fluxes, and ecosystem respiration, via a data fusion approach[42]. The biogenic CO$_2$ exchange, including CO$_2$ exchange from urban green spaces, is parameterized in ASLUM to resolve the interplay between physical environment and biochemical processes. The urban gross primary productivity (GPP$_u$) in a calculation unit is formulated as

$$\text{GPP}_u = f_V \int_0^{\text{LAI}} F_{\text{GPP}}(\text{PAR}, T_{\text{sk}}, [\text{CO}_2], U, \theta) dL, \quad (1)$$

where function $F_{\text{GPP}}$ is the $A_g$-$r_s$-type plant photosynthesis model in ASLUM; $f_V$ is vegetation fraction (-); LAI is the leaf area index (m$^2$ m$^{-2}$); PAR is photosynthetically activated radiation (Wm$^{-2}$); $T_{\text{sk}}$ is leaf skin

temperature (°C); [CO$_2$] is near-surface CO$_2$ concentration level (ppm); $U$ is the near-surface wind speed, and $\theta$ is the normalized soil moisture in urban green spaces (-). The special integral sums leaf-level carbon assimilation rate to canopy-level primary production when considering the light extinction inside of the canopy, defined in Eq. (7) and (8) in ref. 42. In-canyon ecosystem respiration (R$_u$) is calculated as

$$\text{R}_u = f_s F_R(T_{\text{soil}}, \theta, \text{LAI}), \quad (2)$$

where $F_R$ is the temperature-dependent respiration function in ASLUM; $f_s$ is soil fraction; and $T_{\text{soil}}$ is temperature of the surface layer of soil (°C). The detailed formulation of $F_{\text{GPP}}$ and $F_R$ is described in refs. 42,55.

Each calculation unit in ASLUM represents a certain type of urban land surface, with the land use portfolio derived from the WRF configuration. To reflect the heterogeneity within a grid cell, ASLUM further divides each grid into low, medium, and high-density developed fractions, with the GPP$_u$ and R$_u$ calculated as the linear summation of those fractions, as

$$\text{GPP}_u = f_{u,L} \text{GPP}_{u,L} + f_{u,M} \text{GPP}_{u,M} + f_{u,H} \text{GPP}_{u,H}, \quad (3)$$

where the $f_u$ is the urban fractions and subscripts, $L$, $M$, and $H$ represent low, medium, and high-density developed area, respectively. The urban net ecosystem exchange (NEE$_u$) is calculated as

$$\text{NEE}_u = \text{R}_u - \text{GPP}_u. \quad (4)$$

NEE$_u$ is directional. A positive value means a net release of CO$_2$ from urban green spaces, while a negative value means net CO$_2$ sequestration. The input variables to drive ASLUM, including street and ground level meteorological conditions, land use fractions, vegetation properties, are derived from WRF output and external datasets, which are described in the following subsections.

### WRF urban modeling framework

Weather Research and Forecasting (WRF) model is a fully compressible, non-hydrostatic numerical weather prediction and atmospheric

simulation system, which is usually used for regional and global applications[56]. WRF/urban modeling framework[40] includes urban canopy schemes to better represent the three-dimensional morphology and heterogeneity in urban land surface. We adopt WRF-model version 4.0[41] with the Advanced Research WRF dynamical solver as the numerical tool in this study with the single-layer urban canopy scheme enabled. The simulation domain is configured to cover the entire CONUS and its surrounding regions in Canada and Mexico, with a 5-km grid size in horizontal directions and 32 model eta levels in vertical direction. We select a set of customized and well-tested physics options for this set of numerical experiments[36]. We also adopt the spectral analysis nudging option, with the top wave numbers 4 and 3 in x and y directions, respectively. The land cover types and fractions in CONUS are derived from National Land Cover Dataset (NLCD) 2011 at original 9-arc-second spatial resolution and aggregated to match the domain setup. We use the default leaf area index (LAI) data in WRF-model that derives from MODIS land use. The land surface processes are simulated using the unified Noah land surface model (Noah-LSM) for non-urban cells and non-urban portion in urban cells, and the single-layer urban canopy model for urban cells and urban portion in urban cells. Each urban cell is further divided into three categories, with the street canyon geometry defined in WRF v4.0 urban parameter table. WRF uses a "tiling approach" to reflect the surface heterogeneity and calculate the meteorological condition, such as temperature, as the weighted average of urban and non-urban portion within urban cells[40]. The selection of physics and nudging options is based on a previous study over CONUS to evaluate the impact from urban irrigation on surface and air cooling[36]. These configurations were also adopted in refs. 57,58. for the assessment of heat mitigation strategies and pedestrian heat exposure at various spatiotemporal resolutions.

To test the irrigation-induced changes in urban ecosystem services, we perform numerical simulations over three summers (May 1st–Aug 31st) during 2013–2015, and conduct controlled experiments with and without urban irrigation. Irrigation is conducted during 21:00–22:00 local time every day and stops once the soil water content reaches the field capacity. This scheme mimics the typical municipal and recommended residential outdoor use of water[59–61]. As the goal of this study is to illustrate the newly proposed modeling framework and to showcase the impact from irrigation on biogenic $CO_2$ exchange as an example, for the sake of simplicity, no special treatment on irrigation scheme is included. Urban irrigation modifies the soil moisture directly, and affects energy redistribution via surface hydroclimate processes.

At each calculation timestep (30 s), WRF simulates surface level temperatures, air pressure, humidity, incoming solar radiation, soil moisture, etc. and aggregates to hourly outputs. These variables are then used in ASLUM as the meteorological forcings to drive photosynthesis and respiration models. The hourly outputs of $GPP_u$ and $R_u$ are recorded in both irrigated and non-irrigated cases.

## External dataset

Apart from the timeseries of near-surface meteorological conditions from WRF, Carbon Tracker 2019B (CT2019) from Global Monitoring Laboratory (GML) is used to provide background $CO_2$ concentration for this modeling framework[62]. CT2019 gridded dataset contains atmospheric $CO_2$ concentration at different vertical levels at 1 degree over North America with a 3-h temporal interval. The $CO_2$ concentration from CT2019 at the lowest atmospheric level is further interpolated to 5-km grid to match the WRF domain setup, serving as an additional forcing for ASLUM.

Daily mean air temperature measured by ground weather stations in the Global Historical Climatology Network daily (GHCNd) from the National Centers for Environmental Information (NCEI) are used to validate the WRF simulation regarding the fundamental weather conditions[63]. Supplementary Fig. 2a shows urban fractions and the

spatial distribution of the selected 428 stations across the CONUS. Those stations have continuous records throughout the three summers and are located in urban cells in the WRF simulation.

Model validation for urban $GPP_u$ and $R_u$ over large spatial scale (i.e., CONUS) is technically difficult, mainly due to the lack of observation of ecosystem services in urban areas. There are several gridded datasets with wide spatial coverage and moderate spatiotemporal resolution available. Most of the products focus on natural biomes rather than the urban ecosystem. For example, MODIS GPP data (MOD17A2H) excludes urban and built-up areas[64]. FluxCom, based on machine learning methods, integrates FLUXNET site-level observations, satellite remote sensing, and meteorological data[65], but fails to provide estimations over cities because of the very limited number of urban stations in FLUXNET database.

On the other hand, some gridded products provide data in built-up regions, though their underlying algorithms are not dedicated to urban ecosystems. FluxSat, derived using FLUXNET eddy covariance tower data and coincident satellite data (MODIS) via advanced data-driven techniques, extrapolates urban GPP from observations over natural land[66]. With the temporal consistence between measured GPP and solar-induced chlorophyll fluorescence (SIF) in megacities[67], making the products derived from SIF a promising candidate to validate GPP in cities, such as Orbiting Carbon Observatory-2-based SIFGPP (SIFGPP[68]), and urban biogenic $CO_2$ fluxes from SMUrF[24], though SMUrF assumes cloud-free conditions all year around. Vegetation Photosynthesis Model GPP (VPMGPP[69]) uses photosynthetically activated radiation (PAR) data to estimate photosynthesis rate across different biomes and provides gridded data in cities. We select VPMGPP as the reference to compare the $GPP_u$ calculated over the urban cells in this study, as it has a similar type of the underlying algorithm to our model and very good performance against in situ measurement.

Similar situation applies to soil/ecosystem respiration data. The Soil Respiration Database (SRDB) compiles field measurement data reported from literature worldwide[70], and provides respiration rates at sparse locations between 1961 and 2017. In the 10366 reported data points, only 22 were measured in urban area or urban lawns, with 5 of them in CONUS. SRDB provides global annual soil respiration maps in 2004 and 2006 derived from the complied data using two different algorithms[71]. They become the only available spatial gridded productions for the reference of respiration rate. It is noteworthy that the time window of soil respiration maps does not match the simulation period in this study, nor does the respiration-related variable (soil respiration in SRDB versus ecosystem respiration in our study). Other available field data on ecosystem respiration dedicated to urban area can be found in literature and have been used in the modeling process, such as residential sites in Phoenix, AZ[72], and Boston, MA[31], although these campaign datasets are of little value in this study due to their limited spatial coverage.

It noteworthy that although the aforementioned gridded datasets have values over urban area, it does not mean the values are the ground truth for $GPP_u$ or $R_u$. Most of the algorithms do not resolve the meteorological conditions inside of the urban street canyon, and are usually validated using measurement sites outside of urban areas. We select these datasets as a reference for our model based on the data coverage, resolution, and underlying algorithms. The proposed framework in this study aims to disentangle the temporal dynamics of biogenic $CO_2$ exchange. Therefore, we only focus on the match of spatial patterns or the order of magnitude when comparing the model results against the selected gridded data.

## Model validation

Daily mean temperature 2 meters above the ground ($T_{2m}$) from WRF simulation is aggregated from the original hourly output and compare against the measured daily mean air temperature from ground stations

(Supplementary Fig. 2b). The root mean square error (RMSE), defined as

$$RMSE = \sqrt{\frac{\sum (X_{sim} - X_{obs})^2}{n}} \qquad (5)$$

is calculated for each year and over the entire simulation period. During summer months in 2013–2015, the model RMSE values are 2.1 °C, 2.3 °C, and 2.2 °C, respectively. The overall RMSE is 2.2 °C during the entire simulation period. The model accuracy is comparable to a previous study using 135 GHCN weather stations over CONUS[36]. Modeling results from WRF with an RMSE around 2 °C is commonly accepted as accurate. In this case, we conclude the model reproduces reliable meteorological conditions for the subsequent photosynthesis and respiration modeling.

For the validation of gross primary productivity, we compare the daily averaged $GPP_u$ from the model output with VPMGPP gridded data (Supplementary Fig. 3). The result shows general agreement with those gridded data on spatial distribution and magnitude. The model RMSE is 6.5 $gCO_2\, m^{-2}\, d^{-1}$ (1.7 $\mu mol\, m^{-2}\, s^{-1}$), which is comparable to the prior process-based modeling studies on biogenic carbon fluxes[24,73]. To better understand the model bias, we calculate the model mean bias (MB) and mean absolute bias (MAB), as

$$MB = \frac{\sum (X_{sim} - X_{obs})}{n}, \qquad (6)$$

and

$$MAB = \frac{\sum |X_{sim} - X_{obs}|}{n} \qquad (7)$$

The MB and MAB are −3.8 $gCO_2\, m^{-2}\, d^{-1}$ (−1.0 $\mu mol\, m^{-2}\, s^{-1}$) and 4.5 $gCO_2\, m^{-2}\, d^{-1}$ (1.2 $\mu mol\, m^{-2}\, s^{-1}$), respectively. The model bias is highly correlated with vegetation parameters such as leaf area index (LAI) and vegetation coverage ($f_v$) with linear and geometric correlations with LAI and $f_v$, respectively (Supplementary Fig. 4a, b). Though the MAB can reach 17.7 $gCO_2\, m^{-2}\, d^{-1}$ (4.7 $\mu mol\, m^{-2}\, s^{-1}$) in the 1% urban cells with the lowest performance, the model performs well in the vast majority of urban areas with an MAB less than 3.8 $gCO_2\, m^{-2}\, d^{-1}$ (1.0 $\mu mol\, m^{-2}\, s^{-1}$). It is noteworthy that the vegetation parameters used in WRF are estimated from a land cover database (NLCD) and aggregated to the desired spatiotemporal resolution, which potentially contributes to the observed model biases. For example, large discrepancies exist between the vegetation data in WRF and other datasets with high spatiotemporal resolution, such as the LAI and $f_v$ from 10-day 300 m Copernicus Global Land Service (CGLS) data (Supplementary Fig. 4c, d). The magnitude of discrepancies generally increases with the value of LAI and $f_v$, corresponding to the increasing model bias. We also notice there is no meaningful correlation between the bias and other model inputs, indicating the high sensitivity of model outputs to vegetation parameters. Therefore, it is recommended to use high-resolution land cover and vegetation data for future implementations of the model in urban areas[74].

Similar to the estimation of $GPP_u$, the estimation of $R_u$ shows agreement with those gridded data in terms of spatial distribution and magnitude (Supplementary Fig. 5). The modeled respiration rate is noticeably higher than values from SRDB in the eastern CONUS. One main reason is that daily values from SRDB is calculated from the annual total respiration, whereas here we focus on $R_u$ during summer months. Since respiration is sensitive to ambient temperature, it will be much higher in summer months, especially in densely vegetated regions. As mentioned in Section 2.3, the SRDB gridded dataset is derived from point-scale observations

mostly over natural biomes. For data consistency, SRDB intentionally removes experimental data in cities due to their high $CO_2$ efflux[71,75], leading to the possible underestimation of $R_u$ in the urban environment. Also, the gridded map of SRDB estimates respiration rate in 2004 and 2006, which are not the same as the simulation period of this study. This will also contribute to the observed discrepancies.

Overall, the model produces reasonable results with explainable discrepancies as compared with existing datasets, indicating its capability of capturing the dynamics of hydroclimate and plant physiological activities in urban areas.

**Comparison to EV adoption**

To put the irrigation-induced change of $NEE_u$ into context, we compare it with the carbon reduction from the electrification of traffic, which is a popular and major action to reduce anthropogenic emissions. From the estimation of U.S. Department of Energy, a typical gasoline passenger vehicle emits 15.7 $kgCO_2$ per day, while a full electric vehicle (i.e., BEV) emits 3.5 $kgCO_2$ per day. These amount to a saving of 12.2 $kgCO_2$ per day or 4.4 tons of $CO_2$ per year by replacing one gasoline car with a BEV. Note that these emission data only include the on-road energy consumption based on national average annual mileage, fuel efficiency of gasoline cars, and national level power mix[46]. In our simulation, the total urban areas in the lower 48 states of CONUS[76] are $2.7 \times 10^{11}\, m^2$, which is in line with the other statistics ($2.8 \times 10^{11}\, m^2$ from ref. 77). The estimation assumes 15% of the total urban areas in CONUS implement irrigation as modeled. The 0.22 $gCO_2\, m^{-2}\, d^{-1}$ additional release accumulates to 8.9 $\times 10^3$ tons $CO_2$ per day (0.22 $gCO_2\, m^{-2}\, d^{-1} \times 15\% \times 2.7 \times 10^{11}\, m^2 \times 10^{-6}$ ton/g). We further assume the change of $NEE_u$ only happens during warm months as modeled (May 1st–Aug 31st, 123 days), while the vehicle is used all year (365 days). In this case, the total additional release due to irrigation ($1.1 \times 10^6$ tons $CO_2$) offsets the effort of replacing 0.25 million gasoline cars to BEVs. The estimation is very conservative as the calculation may overestimate carbon savings from BEV by neglecting the carbon emissions embedded in EV manufacturing and the construction of charging infrastructures. The calculation may also underestimate the urban area that implementing irrigation considering the urban outdoor water use accounts for one-third to half of the residential water use[78], while the residential lands dominate the urban areas. It is also noteworthy that the estimation will vary across the CONUS depending on the state-level power mix. Similarly, the irrigation-induced change of $NEE_u$ varies across climate zones, which will either sequester or release $CO_2$ from their urban green spaces.

## Data availability

All the datasets used in this study are publicly available: CT2019B data is available at https://gml.noaa.gov/ccgg/carbontracker/. GHCNd dataset is available at https://www.ncei.noaa.gov/products/land-based-station/global-historical-climate-network-daily. VPMGPP dataset at https://doi.org/10.6084/m9.figshare.c.3789814. And SRDBv5 dataset is available at https://daac.ornl.gov/SOILS/guides/SRDB_V5.html. Soil respiration derived from SRDBv3 at: https://daac.ornl.gov/CMS/guides/CMS_Global_Soil_Respiration.html. The processed data generated in this study have been deposited in the Zenodo database under the Creative Commons Attribution 4.0 International license and can be accessed from https://doi.org/10.5281/zenodo.10723633.

## Code availability

The community Weather Research & Forecasting (WRF) model is available at https://github.com/wrf-model/WRF (https://doi.org/10.5065/1dfh-6p97). Other computer codes are available from the authors upon request.

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

## Acknowledgements

This study is based upon work supported by the U.S. National Science Foundation (NSF) under Grant # CBET-2028868 and AGS-2300548 (Z.H.W.).

## Author contributions

Z.-H. Wang and P. Li conceived the idea. Z.-H. Wang acquired the fund and supervised the research. P. Li and C. Wang retrieved and processed the data. P. Li performed the numerical simulations and drafted the manuscript. All authors revised the draft and finalized the manuscript.

## Competing interests

The authors declare no competing interests.
