## [Peer Review File · Nature Communications]

The Potential of Urban Irrigation for Counteracting Carbon-Climate FeedbackReviewers' comments:

Reviewer #1 (Remarks to the Author):

The manuscript entitled “Can Urban Irrigation Counteract Carbon-Climate Feedback?” addresses an interesting and important topic. This research investigates the impact of urban irrigation over green spaces on ambient temperatures and CO₂ exchange across 20 major cities in the contiguous United States using a novel WRF-urban modeling framework. It is found that there are irrigation-induced environmental co-benefits, as well tradeoffs, depending on the local climate of the cities. Results from this study can provide insights for enhanced understanding on the impacts of greenery irrigation, one of the widely adopted management practices on urban local climate and carbon emission, and thereby have important implications for policy makers and managers. Overall, the manuscript is very well written. But I have some major concerns about the validation, and some comments for further improvement.

1) One of my major concerns is about the model validation. I agree with the authors that model validation in urban setting remains challenging as most of the in-situ/ground measurements are typically missing from urban environments. With that being said, I found the validation is not satisfactory. For example, the overall RMSE is 2.23 °C during the entire simulation period, but the observed the difference in air temperature is typically less than 1 °C. Is the difference from irrigation, or maybe simply due to bias/errors in modeling? The results of changes in carbon from the modeling may be more questionable. As stated by the authors, “the regression over the 20 major cities indicates additional 4.15 gCO₂ m⁻² d⁻¹ will be released per 1 °C of air cooling”, the bias in grids with high LAI and f_v, however, can reach as high as 25 g m⁻² d⁻¹. I understand it might be challenging to find ground measured datasets to valid the model for the entire 20 cities. But at least, this shall be done for one or a few cities where data are available.

2) Another concern is about the data. Urban greenspace in cities is high fragmented, characterized by the very large number of, but small-sized green patches. This is particularly true for cities in arid climate such as Phoenix. As noted by the authors, it is highly desirable to use high resolution land cover and vegetation data for implementations of the proposed model in urban areas. I recommend testing the proposed model may be just in one city, or a few cities, and comparing the results with the data used in the current version. This would be a valuable contribution.

3) When discussing the co-benefits and tradeoffs, it might be worth noting the benefits of reduction in carbon emissions from cooling of air temperature due to irrigation.

4) Discussion: The authors may want to address the limitation of this study.

Some specific comments

L95: Please define the term CONUS when it first time appeals.

L183: Is this conclusion consistent to the results from previous case studies?

L417: If the land use fractions and vegetation fractions were already used as inputs in the model,

L448: Would 21:00-22:00 be the time when most of the irrigation occur?

L525: Data of leaf area index and vegetation coverage from the MODIS product in urban areas typically have low accuracy, with lots of uncertainty. If the model bias is highly correlated to these parameters,

then the results can be very problematic.

Reviewer #2 (Remarks to the Author):

General:

This paper explores the interesting topic of how irrigating greenspaces in urban areas might affect both local temperatures and biogenic CO₂ fluxes. The authors contrast the cooling effects of adding water to soil on soil respiration with its impacts on plant CO₂ uptake in urban landscapes. Although this is an interesting and timely topic, I think the fundamental flaw with this work is that it is balancing the cooling effects of irrigation with the potentially increased biogenic CO₂ fluxes. This is problematic because biogenic CO₂ fluxes are not necessarily a “bad” thing as the authors suggest. I would guess that even with a stimulatory effect of irrigation on organic matter decomposition, I would imagine that biogenic fluxes are not a big contributor to a city's net C emissions. Have you calculated this? If so and those increases in soil-derived CO₂ from decomposition are on par with some important fossil fuel CO₂ emitting sectors, that would be an interesting finding that should be reported. Of course, this would need to be compared to increases in GPP to quantify the net effect. In my mind, the cooling effects of irrigation (manifested through increased evapotranspiration) should actually have a beneficial impact on a city's net C emissions because cooling air temperatures will reduce electricity demand for cooling. There is a lot of literature on this topic and I think the authors miss a great opportunity here by not focusing on this key co-benefit.

Also, the results/discussion do not address any differences among plant functional types or tree species (there is considerable variation in how species and PFTs respond to water availability and temperature). I did not notice mention of this in the methods, was this explored? Similarly, it seems that a coarse resolution MODIS product was used for LAI. There is a growing body of literature highlighting the importance of using high-resolution products for urban areas because of the high spatial heterogeneity in land covers. I would imagine that the MODIS product provides an underestimate of LAI in urban areas. Has this been assessed? Similarly, in the methods there is mention of using the NLCD land cover products. I think the 30-m spatial resolution of NLCD is helpful for cities, but typically the land cover types are different intensities of development. How was this NLCD product used with the MODIS LAI to reconcile the distribution and type of vegetation across each city?

As a general comment, I appreciate the need to save space and put all of the methods in the methods section at the end of the paper, but I do think there needs to be some brief overview or description of the approach taken in the main body of the text. As written, I find the results quite difficult to interpret without this and for this style of paper I think it is important to provide enough of a methods overview in the main text for the reader to understand how you came to your results. Overall, I think this is a potentially interesting topic, but I strongly suggest the authors rethink/retool research approach and framing of the paper. There is an opportunity to make a great contribution to science and society here, but to do so I think the authors need to improve their methodology (e.g., spatial resolution of LAI, considering PFTs) and place their results in the context of fossil fuel emissions in cities and how cooling

could reduce these emissions by reducing energy demands for cooling. I hope that my suggestions here and detailed comments below are helpful for future iterations of this work.

Abstract:

L22: Should this be "...more specifically the rise...?"

Main:

L49-51: Isn't soil respiration a larger flux than AnCO₂?

L51-54: Do you mean for cooling purposes? How does this balance lower emissions from reduced heating requirements in cool/cold climate regions?

L57: Should this be "nature-based"?

L61-70: What are the knowledge gaps these studies pointed out?

L74-77: This text misses the fundamental difference between CO₂ emissions from vegetated ecosystems and from fossil fuel combustion for cooling. Biogenic fluxes do not cause a net increase in atmospheric CO₂.

L77-78: I do not quite follow the logic. Even if we assume that CO₂ emissions from soil respiration are a driver of net increases in CO₂ in the atmosphere (which could be the case if decomposition was stimulated), wouldn't GPP also increase and potentially offset at least some of this? Isn't the goal of irrigation to increase plant production?

L80: which continent?

L84: NC has broad readership. As such, I suggest defining what WRF stands for and what this model does? Only a narrow subset of readers will be familiar with this.

L88: How widely adopted is this approach in practice? I am not familiar with any cities doing this at scale with an intention of mitigating the UHI.

L89-92: I agree that there could be important CO₂ implications of increasing the latent heat flux across an urban landscape, however I think the important implication is with reduced energy demand for cooling, not changes in biogenic fluxes. The former is the important component or co-benefit for policymakers to consider. Changes in biogenic fluxes are important for urban CO₂ accounting, but unless I am missing an important detail here changes in biogenic fluxes are not important for a city's net CO₂ emissions (I suspect this is true even if irrigation increases decomposition of soil organic matter).

L96-98: Is there really utility in considering irrigation for arid cities that already experience extreme water limitations and are trying to promote a xeric approach to landscaping? Also, if this was truly a continent-scale analysis, why are you focusing the text here on just a few cities from a rather limited distribution of climate zones?

L118-120: This statement seems to ignore the fact that aboveground plant tissues are considerable sources of CO₂ from respiration.

L120-122: This result seems counter to much of what is known about photosynthesis. Sure, the small reduction in temperature might have an impact on the leaf-level biochemical processes that govern photosynthesis, but lower temperatures could also keep leaf temperatures from exceeding optima for photosynthesis AND increased water availability is also likely to increase photosynthesis, particularly when soils are drier and VPD is higher. Why do you think your results produce this outcome? Similarly, I would imagine that in arid landscapes, increasing soil moisture could actually have a stimulatory effect on soil respiration. Thoughts on why your results are different? (I see you get at some of this later on)

L135: Negative impact on what?

L141: What is the relative change (i.e., % difference) compared to the no irrigation scenario? Also, you express these data with per day as the time variable. What does this equate to over the course of the growing season? Year? In your methods you indicate that model bias is 1-2 orders of magnitude higher than these differences. How might that confound interpretation of the data?

L226: This does not make sense to me. A key goal of irrigating urban systems should be cooling the city and mitigating the UHI. After all, this is what will improve public health and reduce electricity demand for cooling. This recommendation seems flawed to me and if embraced as policy would be counterproductive.

L233-234: I'm not sure anyone actually irrigates when the soil is wet, that would be a waste of money and water resources.

Reviewer #3 (Remarks to the Author):

The authors present a very thorough WRF model simulation of the impact of soil moisture controlled irrigation upon air temperature and net ecosystem exchange of CO₂ in urban green spaces for 12 metropolitan regions of USA. The manuscript is generally well written and clearly the method is innovative and thorough. However, there is a central concern with how the term 'Trade off' is being communicated in this context. The cooling benefit and CO₂ exchange in these cities are effectively being viewed as being equally important – but they are not. The cooling benefit is considerable, targeted and meaningful, whereas the CO₂ exchange increase is negligible and inconsequential when global radiative forcing is considered and that is the only way to view CO₂ emissions – globally. The authors must place the 0.22 g m⁻² d⁻¹ in context, a global context, because we can all place the 0.26 °C in context, a global or city context. To do this, the sum total increase in CO₂ emissions should be compared to annual anthropogenic emissions in CONUS for examples – not that will give researchers and decision-makers an opportunity to assess whether this is a 'trade-off' to worry about.

Don't get me wrong, I think this is a great study. But by spinning this to show that with irrigation you can get an important cooling benefit for very little concomitant increase in CO₂ emission from urban vegetation – now that is an impactful study – not just an academic one.

Another more minor issue is that there has been no attempt to consider the CO₂ exchange associated with management of green spaces, even though an initial premise of the study was increase GHG emissions associated with management of building spaces for cooling.

Please find below comments made as I read through the manuscript.

Abstract

L24 The abstract does not clearly define the mechanism of positive climate-carbon feedback in the abstract. I assume (have to) that the authors refer to higher temperatures leading to greater building air conditioning use? and therefore greater GHG emissions associated with energy production to run A/Cs, or plant and soil respiration, or both/all?

L26 Please provide in the abstract an indication of how (method) you investigate the impact of green space irrigation on temperatures and CO₂ exchange". Are you modelling, measuring or meta-analysing?

L28 what is this "irrigation-induced environmental co-benefit" of which you write? Is it perhaps related

to cooling – if so, then say that as ‘environment co-benefit’ could be many things.

L31 Respiration of what? soil, humans, plants? Remember the reader has not read the paper yet, the abstract must be intelligible on its own.

L40 delete ‘the’

L42 changes in global climate rather than global climate changes?

L50 the way this sentence is currently worded it is not strictly true - AnCO₂ does not represent the largest carbon flux to the atmosphere – it does, however, represent the greatest carbon flux that drives the increase in radiative forcing in the atmosphere.

L47 to 56 – this is a logic breakdown – fossil fuels are not burned in buildings to run air conditioners – they are burned in power stations, away from urban areas, to create electricity that is used to run air conditioners in urban areas.

L58 I would strongly disagree that nature-based solutions have been extensively adopted to counteract environmental impacts concomitant with global urbanization. Yes, they have been studied and written about BUT they have not been extensively adopted, especially not urban greening, as many western towns and cities are going backwards in urban green cover and urban forest cover.

L82 – rather limited to a few cities? – Citation 24 studied 100 global cities?

L86 and L91 pick one term. Either thermal-carbon OR heat-carbon. Then use throughout.

L95 CONUS? Define the first time you use.

L95 What are you irrigating in these 12 metropolitan regions? The whole city? All urban green space? Public and private green space? How much are you irrigating? What time of day are you irrigating (at night)? How does WRF simulate irrigation? This may well be in the methods section – but please tell the reader here in a just a few words some of these things – they want to know.

Figure 2, panel a) does not indicate higher background CO₂ concentration. Also, higher than what? indicate this.

Figures 4 and 5 – are both of these figures really necessary? Perhaps supplementary – it is the NEE that you can talk to when you get to tradeoffs.

L134 – okay you present the CO₂ exchange impact in g CO₂ m⁻² d⁻¹, which is fine BUT not very informative to the reader. It is very easy for all readers to place the cooling impact of 0.26 °C from urban greenspace irrigation in context with say predicted global warming of 1.5°C. However, what you need to do is place this CO₂ exchange in context with global C exchange (daily or annually). L182 - The reason being is that this so-called trade-off, I suspect, is not really a trade-off to worry about at all. The cooling benefit is considerable, targeted and meaningful, whereas the CO₂ exchange increase is negligible and inconsequential when global radiative forcing is considered and that is the only way to view CO₂ emissions – globally. If I am wrong, and even if I am not, please, place this irrigation impact on CO₂ exchange in context for the reader. This appears to be a very detailed academic modelling exercise – but please indicate the real impact of these CO₂ flux in a global C flux context.

Another issue is that there has been no attempt to consider the CO₂ exchange associated with management of green spaces, even though an initial premise of the study was increase GHG emissions associated with management of building spaces for cooling. There are several studies that have indicated the CO₂ sequestration by urban vegetation is offset by the combined impact of management events and embedded energy associated and respiration flux.

L223 unintended carbon release is so small in comparison to the associated cooling benefit – this is at the heart of the misleading nature of this study.

L226 'avoid excessive air cooling'? really?

L245 – 256 I completely agree – it is very complex and models are inadequate to simulate these process interactions yet, let alone simple irrigation-temperature-carbon interactions. As such, this study must not better represent and communicate the impact of irrigation of CO₂ exchange in a global radiative forcing context, otherwise this study is just preventing us from getting to good urban water and green space decision making sooner.

Point-by-point response to referee comments

We sincerely thank all three reviewers and the editor for their constructive feedback and help in improving the quality of this manuscript. Below are detailed responses to the comments. All changes and clarifications were included in the revised manuscript.

Reviewer #1 (Remarks to the Author):

The manuscript entitled “Can Urban Irrigation Counteract Carbon-Climate Feedback?” addresses an interesting and important topic. This research investigates the impact of urban irrigation over green spaces on ambient temperatures and CO₂ exchange across 20 major cities in the contiguous United States using a novel WRF-urban modeling framework. It is found that there are irrigation-induced environmental co-benefits, as well tradeoffs, depending on the local climate of the cities. Results from this study can provide insights for enhanced understanding on the impacts of greenery irrigation, one of the widely adopted management practices on urban local climate and carbon emission, and thereby have important implications for policy makers and managers. Overall, the manuscript is very well written. But I have some major concerns about the validation, and some comments for further improvement.

We thank the reviewer’s positive feedback. Please find the point-to-point responses in the following text.

1) One of my major concerns is about the model validation. I agree with the authors that model validation in urban setting remains challenging as most of the in-situ/ground measurements are typically missing from urban environments. With that being said, I found the validation is not satisfactory. For example, the overall RMSE is 2.23 °C during the entire simulation period, but the observed the difference in air temperature is typically less than 1 °C. Is the difference from irrigation, or maybe simply due to bias/errors in modeling? The results of changes in carbon from the modeling may be more questionable. As stated by the authors, “the regression over the 20 major cities indicates additional 4.15 gCO₂ m⁻² d⁻¹ will be released per 1 °C of air cooling”, the bias in grids with high LAI and f_v, however, can reach as high as 25 g m⁻² d⁻¹. I understand

it might be challenging to find ground measured datasets to valid the model for the entire 20 cities. But at least, this shall be done for one or a few cities where data are available.

The high uncertainty in the estimation and modeling is a key point we underscored in this study. We also addressed that there is no “ground truth” for urban biogenic CO₂ fluxes for the moment. The validation, however, is based on the current availability of the spatial gridded data, which estimates from various approaches and with different underlying assumptions. For example, the latest SMUrF urban dataset (Wu et al. 2021) assumes cloud-free conditions all year around. VPMGPP used MODIS data, which sometimes has substantial errors and biases (Wang et al. 2017). The bias between SMUrF and VPMGPP is also large in urban regions. The model bias we disclosed in this study represents the *difference* between VPMGPP and our simulation (reference case). Therefore, we cannot conclude that our results are “accurate or not” based on the products with their own uncertainties. It is noteworthy that, in the revision, we evaluated our model performance using root mean squared error (RMSE) in addition to the mean absolute bias (MAB) used previously. The model RMSE is 6.5 gCO₂ m⁻² d⁻¹ (1.7 μmol m⁻² s⁻¹) against VPMGPP, which is comparable to the prior process-based modeling studies on biogenic carbon fluxes at daily (Wu et al., 2021) and annual scale (Madani et al., 2017). The previously reported “as high as 25 gCO₂ m⁻² d⁻¹” is the largest bias observed in the simulation. Model error in most area is significantly lower than this value. The figure below shows the distribution of model bias as MAB.

The validation also illustrates that our simulation can reflect the dynamics of thermal and carbon environments over a large region, which implies our model is *precise*. The scientific

question we would like to investigate is “what are the impacts of urban irrigation”; therefore, the study focused more on the change before and after irrigation. Since the model is precise, the general trends (dT_{2m} from CONUS, and $dNEE_u/dT$ from 20 cities) will not be largely affected by model accuracy. Once there were reliable data serving as the “ground truth”, the modeling framework could be re-calibrated to be more *accurate*. Unfortunately, to the best of our knowledge, there is no known, well-validated, and comparable dataset to use. There are only few point-scale measurements in some cities, which are already discussed in the manuscript. The large uncertainties showed in this study imply the notable scientific gap in estimating urban biogenic processes.

To improve the manuscript, we include a short discussion in the revision on the limitation of this study per the reviewer’s suggestion.

Reference:

- Wu, D., Lin, J. C., Duarte, H. F., Yadav, V., Parazoo, N. C., Oda, T., & Kort, E. A. (2021). A model for urban biogenic CO₂ fluxes: Solar-Induced Fluorescence for Modeling Urban biogenic Fluxes (SMUrF v1). *Geoscientific Model Development*, 14(6), 3633-3661. <http://doi.org/10.5194/gmd-14-3633-2021>
- Wang, L., Zhu, H., Lin, A., Zou, L., Qin, W., & Du, Q. (2017). Evaluation of the Latest MODIS GPP Products across Multiple Biomes Using Global Eddy Covariance Flux Data. *Remote Sensing*, 9(5), 418. <https://doi.org/10.3390/rs9050418>
- Madani, N., Kimball, J. S., & Running, S. W. (2017). Improving global gross primary productivity estimates by computing optimum light use efficiencies using flux tower data. *Journal of Geophysical Research: Biogeosciences*, 122(11), 2939-2951. <http://doi.org/https://doi.org/10.1002/2017JG004142>
-

2) Another concern is about the data. Urban greenspace in cities is high fragmented, characterized by the very large number of, but small-sized green patches. This is particularly true for cities in arid climate such as Phoenix. As noted by the authors, it is highly desirable to use high resolution land cover and vegetation data for implementations of the proposed model in urban areas. I recommend testing the proposed model may be just in one city, or a few cities,

and comparing the results with the data used in the current version. This would be a valuable contribution.

We totally agree with the reviewer's comments and thank his/her suggestion. In fact, we are currently incorporating the high-resolution data (300-m biomass maps and 30-m land cover data) with the modeling framework proposed in this study over the large metropolitan regions, such as the greater Chicago area (Li et al., 2023). While in this study, we emphasized the *spatial variation* of irrigation-induced impacts over CONUS. While comparison among cities across the entire CONUS is more meaningful since this can reveal the impacts of various controlling factors on urban biogenic carbon fluxes, simulating meteorological and biogeochemical dynamics over CONUS at a very high resolution is not practical. In the development phase of this study, we have tested CONUS-scale simulations at multiple spatial resolutions. Running WRF-UBC for CONUS at resolution finer than 5km is extremely time consuming and demands tremendous computational resources. On the other hand, 5km is already much finer than common Earth System Models (ESM) runs, which are widely used in IPCC assessment reports.

In the revision, we include a short discussion on the limitation of this study and a brief discussion on the potential future work on the high-resolution modeling over specific cities.

Reference

Li, P., Sharma, A., Wang, Z.-H., & Wuebbles, J.D. (2023) Assessing impacts of environmental perturbations on urban biogenic carbon exchange in the Chicago region. *Authorea*.
<http://doi.org/10.22541/essoar.167898495.56632750/v1>

3) When discussing the co-benefits and tradeoffs, it might be worth noting the benefits of reduction in carbon emissions from cooling of air temperature due to irrigation.

In Figure 2, we showed the pathways that affecting the biogenic CO₂ exchange. It turned out that air cooling is not good for carbon reduction as it reduces GPP (Fig. 2b). But air cooling is always associated with soil cooling, and soil cooling reduce CO₂ emission. Though they follow the different underlying mechanisms, for most of the time, air cooling helps reduce carbon emission (but with exceptions). In this case, we highlighted the mechanisms behind the apparent outcomes so that the different regions/cities should consider the location-aware solutions. Therefore, we did not specifically claim the co-benefits of air cooling and carbon

reduction. Instead, soil cooling is always good for carbon reduction. Also, soil cooling was more discussed as the air cooling due to irrigation is less reported in the other literature.

4) Discussion: The authors may want to address the limitation of this study.

To improve the manuscript, we include a short discussion in the revision on the limitation of this study per the reviewer's suggestion. The primary limitation is caused by the constraints of data availability and computational resources, including (1) limited studies on urban ecosystem services, (2) the scarcity of measurement of biogenic emissions in urban areas, and (3) the lack of comprehensive modeling framework for urban dynamics inclusive of CO₂ exchange.

Some specific comments

L95: Please define the term CONUS when it first time appeals.

In the revision, we defined "CONUS" at its first appearance.

L183: Is this conclusion consistent to the results from previous case studies?

A modeling study (Li and Wang 2021) showed similar results. Observation-wise, the irrigated scenario needs a well-controlled experiment to replicate. There are two publications explicitly addressed the impact of irrigation on urban NEE in Boston, MA, and Phoenix AZ (Decina et al. 2016 and Kindler et al. 2022). Both studies indicated irrigation will significantly increase soil respiration, thus affect the net CO₂ exchange. Their conclusions aligned with our modeling results. Unfortunately, these campaign datasets are of little value due to their limited spatial coverage. The observations over biogenic sector in urban environment are very rare and have a very limited spatiotemporal coverage. Therefore, in the last paragraph, we encourage and appeal the research community for the effort on urban observation and data synthesis over biogenic sectors.

References:

Li, P., & Wang, Z.-H. (2021). Environmental co-benefits of urban greening for mitigating heat and carbon emissions. *Journal of Environmental Management*, 293, 112963.

<http://doi.org/10.1016/j.jenvman.2021.112963>

Decina, S. M., Hutyra, L. R., Gately, C. K., Getson, J. M., Reinmann, A. B., Short Gianotti, A. G., & Templer, P. H. (2016). Soil respiration contributes substantially to urban carbon fluxes in the greater Boston area. *Environmental Pollution*, 212, 433-439.

<http://doi.org/10.1016/j.envpol.2016.01.012>

Kindler, M., Vivoni, E. R., Pérez-Ruiz, E. R., & Wang, Z. (2022). Water conservation potential of modified turf grass irrigation in urban parks of Phoenix, Arizona. *Ecohydrology*, 15(3), e2399. <http://doi.org/https://doi.org/10.1002/eco.2399>

L417: If the land use fractions and vegetation fractions were already used as inputs in the model.

Yes. Both were used as inputs.

L448: Would 21:00-22:00 be the time when most of the irrigation occur?

Yes. Despite irrigation schedules varies a lot over the CONUS, urban irrigation usually happens at late night to mid-night. It is also recommended to irrigate after sunset and before sunrise to minimize water loss due to evapotranspiration. The exact time for irrigation at night will not affect the result by much. It noteworthy that the irrigation scenario is primarily designed to demonstrate the impact on urban biogenic CO₂ exchange from irrigation. Though we aimed to align the numerical experiment with the real case as close as possible, it is extremely difficult to replicate the potential diverse irrigation behaviors that vary across the country.

L525: Data of leaf area index and vegetation coverage from the MODIS product in urban areas typically have low accuracy, with lots of uncertainty. If the model bias is highly correlated to these parameters, then the results can be very problematic.

The WRF urban modeling framework handles LAI data in a tiling approach that combines the information from MODIS LAI, look-up table, urban density, and land use types (Chen et al. 2002; 2011). This information is aggregated to the desired spatial resolution in the model run. We compared the WRF LAI to CGLS data and showed the results in Figure S4d. For most areas, the LAI values are aligned with a mean bias of $-0.2 \text{ m}^2\text{m}^{-2}$ compared to CGLS LAI. Since WRF-UCM has been widely used in urban climate research without apparent issues, we adopted this framework and its default LAI data. For studies at CONUS scale, it is computationally expensive to use alternative data with high resolution. Meanwhile, we

conducted city level assessment (much smaller domain as limited by computational cost) where high resolution land cover and biomass data are feasible in a model with a much higher spatial resolution (Li et al., 2023).

References:

- Chen, F., & Dudhia, J. (2001). Coupling an Advanced Land Surface–Hydrology Model with the Penn State–NCAR MM5 Modeling System. Part II: Preliminary Model Validation. *Monthly Weather Review*, 129(4), 587-604. [http://doi.org/10.1175/1520-0493\(2001\)129<0587:Caalsh>2.0.Co;2](http://doi.org/10.1175/1520-0493(2001)129<0587:Caalsh>2.0.Co;2)
- Chen, F., Kusaka, H., Bornstein, R., Ching, J., Grimmond, C. S. B., Grossman-Clarke, S., . . . Zhang, C. (2011). The integrated WRF/urban modelling system: development, evaluation, and applications to urban environmental problems. *International Journal of Climatology*, 31(2), 273-288. <http://doi.org/10.1002/joc.2158>
- Li, P., Sharma, A., Wang, Z.-H., & Wuebbles, J.D. (2023) Assessing impacts of environmental perturbations on urban biogenic carbon exchange in the Chicago region. *Authorea*. <http://doi.org/10.22541/essoar.167898495.56632750/v1>
-

Reviewer #2 (Remarks to the Author):

General:

This paper explores the interesting topic of how irrigating greenspaces in urban areas might affect both local temperatures and biogenic CO₂ fluxes. The authors contrast the cooling effects of adding water to soil on soil respiration with its impacts on plant CO₂ uptake in urban landscapes. Although this is an interesting and timely topic, I think the fundamental flaw with this work is that it is balancing the cooling effects of irrigation with the potentially increased biogenic CO₂ fluxes. This is problematic because biogenic CO₂ fluxes are not necessarily a “bad” thing as the authors suggest. I would guess that even with a stimulatory effect of irrigation on organic matter decomposition, I would imagine that biogenic fluxes are not a big contributor to a city's net C emissions. Have you calculated this? If so and those increases in soil-derived CO₂ from decomposition are on par with some important fossil fuel CO₂ emitting sectors, that would be an interesting finding that should be reported. Of course, this would need to be compared to increases in GPP to quantify the net effect. In my mind, the cooling effects of irrigation (manifested through increased evapotranspiration) should actually have a beneficial impact on a city's net C emissions because cooling air temperatures will reduce electricity demand for cooling. There is a lot of literature on this topic and I think the authors miss a great opportunity here by not focusing on this key co-benefit.

Thanks for the insightful comments. We would like to clarify that we did not intend to suggest biogenic CO₂ exchange is necessarily a *bad* thing. Instead, our investigation suggests that by understanding the interplays between urban microclimate, anthropogenic inference, and plant/soil functions, we can manage urban green spaces for better climate change adaptation and mitigation in cities. As the title indicates, we would like to answer, “can urban irrigation counteract the carbon-climate feedback”, instead of “how much CO₂ is released from ecosystem respiration” (though the quantification is necessary to answer the title question).

Second, we focused on the quantification of the **change** in urban NEE before and after irrigation is 0.22 gCO₂ m⁻²d⁻¹. In the global context: the latest estimate of the posterior 5-year annual mean global atmospheric CO₂ growth rate was 5.35 PgC yr⁻¹ (Jin et al., 2021), which converts to 0.41 gCO₂ m⁻²d⁻¹ over global land area. It is clear that the change of CO₂ flux due to irrigation is significant. In terms of cities emission portfolio, the statement “*biogenic C fluxes are*

not a big contributor to a city's net carbon emission” is not always true. Previous case studies found soil respiration in residential area is comparable to traffic emissions during summer months (Decina et al. 2016), and the increase is directly related to lawn irrigation. In addition, the change in biogenic emission is significant when comparing to the ongoing carbon reduction effort from the anthropogenic sector. It is more reasonable to compare the change to change. In the revision, we included a calculation to provide a background for the irrigation-induced change by comparing it to the carbon reduction from changing gasoline car to electric cars, indicating the irrigation-induced NEE change offsets the carbon reduction from replacing 0.25 million gasoline cars to the cleanest battery EV cars.

Third, we agree that “*there is a lot of literature on this topic (of cooling-energy benefit)*”, and our research group has made contribution on this topic (e.g., Wang et al. 2016). Nevertheless, the current study is devoted to a much under-explored topic of study on the co-benefit of cooling and carbon mitigation from urban green spaces. This is enabled by our recent development of the state-of-the-art Arizona Single Layer Urban canopy Model (ASLUM v4.0) featuring the holistic parameterization of biogenic, abiotic, and anthropogenic CO₂ exchange in the urban environment (Li and Wang, 2020). As many cities plan to implement nature-based solutions to reduce carbon emission without knowing the underlying tradeoffs, our modeling framework contribute to the decision-making and the actionable science. The current study on biogenic CO₂ exchange and its interplay with urban thermal environment bridges the critical knowledge gap to further our fundamental understanding in the much-needed, but yet under-explored, quantification of heat-carbon feedback in the repercussion of climate change and energy-carbon-water trade-offs.

References:

- Decina, S. M., Hutyra, L. R., Gately, C. K., Getson, J. M., Reinmann, A. B., Short Gianotti, A. G., & Templer, P. H. (2016). Soil respiration contributes substantially to urban carbon fluxes in the greater Boston area. *Environmental Pollution*, 212, 433-439.
<http://doi.org/10.1016/j.envpol.2016.01.012>
- Li, P., & Wang, Z.H. (2020). Modeling carbon dioxide exchange in a single-layer urban canopy model. *Building and Environment*, 184, 107243.
<https://doi.org/10.1016/j.buildenv.2020.107243>

Jin, Z., Tian, X., Han, R., Fu, Y., Li, X., Mao, H., & Chen, C. (2021). A global CO₂ flux dataset (2015–2019) inferred from OCO-2 retrievals using the Tan-Tracker inversion system. *Earth System Science Data Discussion*, 2021, 1-40. <https://doi.org/10.5194/essd-2021-210>

Wang, Z.-H., Zhao, X., Yang, J., & Song, J. (2016). Cooling and energy saving potentials of shade trees and urban lawns in a desert city. *Applied Energy*, 161, 437-444. <http://doi.org/https://doi.org/10.1016/j.apenergy.2015.10.047>

Also, the results/discussion do not address any differences among plant functional types or tree species (there is considerable variation in how species and PFTs respond to water availability and temperature). I did not notice mention of this in the methods, was this explored? Similarly, it seems that a coarse resolution MODIS product was used for LAI. There is a growing body of literature highlighting the importance of using high-resolution products for urban areas because of the high spatial heterogeneity in land covers. I would imagine that the MODIS product provides an underestimate of LAI in urban areas. Has this been assessed? Similarly, in the methods there is mention of using the NLCD land cover products. I think the 30-m spatial resolution of NLCD is helpful for cities, but typically the land cover types are different intensities of development. How was this NLCD product used with the MODIS LAI to reconcile the distribution and type of vegetation across each city?

Our latest urban land surface model, ASLUM v4.0, incorporates different biogenic functions and urban vegetation species (Li and Wang, 2020; 2021), including different parameterizations for C3 and C4 plants, vegetation height, location on the streets, etc. The primary environmental factors that control the growth of urban plants have been well-considered in the modeling processes. In comparison, in the *status quo* of mesoscale urban climate modeling, the distinction of individual species or different plant functional types remain largely missing. This is mainly due to the high heterogeneity of the urban surfaces and morphology. We agree that resolving urban climate dynamics needs high resolution land cover data. But for physical-based numerical models over continental scale, incorporating the details of vegetation will increase the computational cost exponentially, and eventually makes the numerical experiment non-practical. One needs to balance the domain coverage and resolution. In this study, we adopted three urban categories (defined by NLCD) within the 5 km calculation cell

over the entire CONUS urbanized areas and run the simulation over three summers (369 days in total). The parameterization can better reflect urban heterogeneity than the default tiling approach commonly used in numerical weather prediction models. This intensive calculation task was performed on supercomputers in National Center for Atmospheric Research and Arizona State University. We are currently working on the simulation over specific metro regions (250km × 250km domain) to achieve higher resolution (~300m) (Li et al., 2023). Even with the reduced domain size, achieving street-level resolution is challenging and beyond the status quo of urban climate studies.

The WRF urban modeling framework (Chen et al. 2001; 2011) handles LAI data in a tiling approach that combines the information from MODIS LAI, look-up table, urban density, and land use types. This information is aggregated to the desired spatial resolution in the model run. Since WRF urban modeling framework has been widely used in urban climate research without apparent issues, we adopted this framework and compared the WRF LAI to CGLS data and showed the results in Figure S4b. For most areas, the LAI values are aligned. The overall mean bias is $-0.2 \text{ m}^2\text{m}^{-2}$ compared to CGLS LAI, equivalent to a ~0.6% relative difference.

The NLCD products are used in a similar way. WRF urban modeling framework aggregates 30-m data to desired spatial resolution for model run. In fact, NLCD distinguishes development intensities and classified urban land use types. The dataset also provides fractions of canopy coverage and impervious surfaces, therefore has the ability to interpret the biomass density and distribution over the urban region. As WRF-urban is a well-established method for urban modeling studies, the parameterization and treatment process of biomass is quite standard and is well documented in the user manual of WRF (Skamarock et al. 2019; Chen et al. 2011).

References:

- Li, P., & Wang, Z.-H. (2020). Modeling carbon dioxide exchange in a single-layer urban canopy model. *Building and Environment*, 184, 107243.
<http://doi.org/10.1016/j.buildenv.2020.107243>
- Li, P., & Wang, Z.-H. (2021). Uncertainty and sensitivity analysis of modeling plant CO₂ exchange in the built environment. *Building and Environment*, 189, 107539.
<http://doi.org/10.1016/j.buildenv.2020.107539>

- Li, P., Sharma, A., Wang, Z.-H., & Wuebbles, J.D. (2023) Assessing impacts of environmental perturbations on urban biogenic carbon exchange in the Chicago region. *Authorea*. <http://doi.org/10.22541/essoar.167898495.56632750/v1>
- Chen, F., & Dudhia, J. (2001). Coupling an Advanced Land Surface–Hydrology Model with the Penn State–NCAR MM5 Modeling System. Part II: Preliminary Model Validation. *Monthly Weather Review*, 129(4), 587-604. [http://doi.org/10.1175/1520-0493\(2001\)129<0587:Caalsh>2.0.Co;2](http://doi.org/10.1175/1520-0493(2001)129<0587:Caalsh>2.0.Co;2)
- Chen, F., Kusaka, H., Bornstein, R., Ching, J., Grimmond, C. S. B., Grossman-Clarke, S., . . . Zhang, C. (2011). The integrated WRF/urban modelling system: development, evaluation, and applications to urban environmental problems. *International Journal of Climatology*, 31(2), 273-288. <http://doi.org/10.1002/joc.2158>
- Skamarock, W. C., Klemp, J. B., Dudhia, J., Gill, D. O., Barker, D. M., Duda, M. G., . . . Powers, J. G. (2019). A description of the advanced research WRF version 4. <http://doi.org/10.6084/m9.figshare.7369994.v4>
-

As a general comment, I appreciate the need to save space and put all of the methods in the methods section at the end of the paper, but I do think there needs to be some brief overview or description of the approach taken in the main body of the text. As written, I find the results quite difficult to interpret without this and for this style of paper I think it is important to provide enough of a methods overview in the main text for the reader to understand how you came to your results.

We thank the reviewer's feedback. The main text has been revised to include an overview of the method in the main text.

Overall, I think this is a potentially interesting topic, but I strongly suggest the authors rethink/retool research approach and framing of the paper. There is an opportunity to make a great contribution to science and society here, but to do so I think the authors need to improve their methodology (e.g., spatial resolution of LAI, considering PFTs) and place their results in the context of fossil fuel emissions in cities and how cooling could reduce these emissions by reducing energy demands for cooling. I hope that my suggestions here and detailed comments below are helpful for future iterations of this work.

We thank the reviewer's suggestion. In the revision, we modified the text to improve the clarity of the manuscript and put our results in the national and global context for better readability. Please find the revised manuscript attached.

Abstract:

L22: Should this be "...more specifically the rise..."?

We corrected this typo in revision.

Main:

L49-51: Isn't soil respiration a larger flux than AnCO₂?

Yes, soil respiration is a larger terrestrial carbon source than AnCO₂. But it is not necessarily a driver for climate change in the Anthropocene, as soil respiration happens even before industrialization. We also understand global warming may alter the rate of soil respiration and has complex interplays (Carey et al. 2016; Wang et al. 2018; Lei et al. 2021).

In this sentence, we meant to say in the potential drivers to climate change, AnCO₂ is arguably the largest carbon flux. The text has been modified to improve the clarity.

References:

- Carey, J. C., Tang, J., Templer, P. H., Kroeger, K. D., Crowther, T. W., Burton, A. J., . . . Tietema, A. (2016). Temperature response of soil respiration largely unaltered with experimental warming. *Proceedings of the National Academy of Sciences of USA*, 113(48), 13797-13802. <http://doi.org/10.1073/pnas.1605365113>
- Wang, T., Liu, D., Piao, S., Wang, Y., Wang, X., Guo, H., . . . Zhao, Y. (2018). Emerging negative impact of warming on summer carbon uptake in northern ecosystems. *Nature Communications*, 9(1), 5391. <http://doi.org/10.1038/s41467-018-07813-7>
- Lei, N., & Han, J. (2020). Effect of precipitation on respiration of different reconstructed soils. *Scientific Reports*, 10(1), 7328. <http://doi.org/10.1038/s41598-020-63420-x>
-

L51-54: Do you mean for cooling purposes? How does this balance lower emissions from reduced heating requirements in cool/cold climate regions?

Yes, we refer to the electricity consumed by buildings. We understand the reviewer's refutation that global warming may be beneficial for cold regions as people there will save energy from less space heating. Indeed, in cities with building energy use dominated by space heating, emissions might decrease in a warmer future. However, for cities with energy use dominated by cooling, the emission reduction induced by less space heating will likely be completely offset by the increased cooling demand. Note these discussions are not the theme of this study. We meant to provide a global background in this sentence. The text has been modified for better clarity.

L57: Should this be "nature-based"?

We corrected this typo in revision.

L61-70: What are the knowledge gaps these studies pointed out?

In the revision, we added a brief summary of the knowledge gaps identified from these studies.

Revised text:

Despite the tremendous research effort, quantifying the comprehensive impact of urban greening on Earth's climate, especially on heat and CO₂ emissions, remains challenging to researchers given the large spatio-temporal uncertainties of biogenic CO₂ exchange (i.e., CO₂ uptake and release via photosynthesis and respiration processes)²¹⁻²³ and the very limited urban observations on biogenic sectors at global scale²⁴. Recent reviews also found significant gaps in implementing nature-based solutions, such as the uncertainties on the effectiveness of the solution²⁵ and lack of public awareness and trustworthy information²⁶, which cause operational barriers for stakeholders.

L74-77: This text misses the fundamental difference between CO₂ emissions from vegetated ecosystems and from fossil fuel combustion for cooling. Biogenic fluxes do not cause a net increase in atmospheric CO₂.

Previous studies at various spatial scales have demonstrated that there are additional CO₂ releasing from biogenic sectors, especially from soil respiration, due to warming and/or

anthropogenic inferences (Decina et al. 2016; Xue and Tang 2017; Jian et al. 2018; Nyberg et al. 2020; Lei et al. 2021; Apostolakis et al. 2022; Kindler et al. 2022; Swails et al. 2022). Moreover, several studies on urban lawns, turf grass, or decayed landscape have also reported net CO₂ emissions from urban green spaces (Allaire et al. 2008; Miller et al. 2020; Hundertmark et al. 2021), which directly contradicts the reviewers claim that “*Biogenic fluxes do not cause a net increase in atmospheric CO₂*”.

Though in general and on global average, it is unlikely the biogenic fluxes lead to a *net increase* of CO₂, there is a notable *net decrease* of CO₂ sink power (Piao et al. 2017; Wang et al. 2018; Tang et al. 2022), which is already alarming before the catastrophic “net increase” happens.

References:

- Decina, S. M., Hutrya, L. R., Gately, C. K., Getson, J. M., Reinmann, A. B., Short Gianotti, A. G., & Templer, P. H. (2016). Soil respiration contributes substantially to urban carbon fluxes in the greater Boston area. *Environmental Pollution*, 212, 433-439.
<http://doi.org/10.1016/j.envpol.2016.01.012>
- Xue, H., & Tang, H. (2018). Responses of soil respiration to soil management changes in an agropastoral ecotone in Inner Mongolia, China. *Ecology and Evolution*, 8(1), 220-230.
<http://doi.org/10.1002/ece3.3659>
- Jian, J., Steele, M. K., Day, S. D., & Thomas, R. Q. (2018). Future global soil respiration rates will swell despite regional decreases in temperature sensitivity caused by rising temperature. *Earth's Future*, 6(11), 1539-1554.
<http://doi.org/https://doi.org/10.1029/2018EF000937>
- Nyberg, M., & Hovenden, M. J. (2020). Warming increases soil respiration in a carbon-rich soil without changing microbial respiratory potential. *Biogeosciences*, 17(17), 4405-4420.
<http://doi.org/10.5194/bg-17-4405-2020>
- Lei, J., Guo, X., Zeng, Y., Zhou, J., Gao, Q., & Yang, Y. (2021). Temporal changes in global soil respiration since 1987. *Nature Communications*, 12(1), 403.
<http://doi.org/10.1038/s41467-020-20616-z>
- Apostolakis, A., Schöning, I., Michalzik, B., Klaus, V. H., Boeddinghaus, R. S., Kandeler, E., . . . Schrupf, M. (2022). Drivers of soil respiration across a management intensity

- gradient in temperate grasslands under drought. *Nutrient Cycling in Agroecosystems*, 124(1), 101-116. <http://doi.org/10.1007/s10705-022-10224-2>
- Kindler, M., Vivoni, E. R., Pérez-Ruiz, E. R., & Wang, Z. (2022). Water conservation potential of modified turf grass irrigation in urban parks of Phoenix, Arizona. *Ecohydrology*, 15(3), e2399. <http://doi.org/https://doi.org/10.1002/eco.2399>
- Swails, E. E., Ardón, M., Krauss, K. W., Peralta, A. L., Emanuel, R. E., Helton, A. M., . . . Ward, S. (2022). Response of soil respiration to changes in soil temperature and water table level in drained and restored peatlands of the southeastern United States. *Carbon Balance and Management*, 17(1), 18. <http://doi.org/10.1186/s13021-022-00219-5>
- Hundertmark, W. J., Lee, M., Smith, I. A., Bang, A. H. Y., Chen, V., Gately, C. K., . . . Hutryra, L. R. (2021). Influence of landscape management practices on urban greenhouse gas budgets. *Carbon Balance and Management*, 16(1), 1. <http://doi.org/10.1186/s13021-020-00160-5>
- Miller, J. B., Lehman, S. J., Verhulst, K. R., Miller, C. E., Duren, R. M., Yadav, V., . . . Sloop, C. D. (2020). Large and seasonally varying biospheric CO₂ fluxes in the Los Angeles megacity revealed by atmospheric radiocarbon. *Proceedings of the National Academy of Sciences of USA*, 117(43), 26681-26687. <http://doi.org/doi:10.1073/pnas.2005253117>
- Allaire, S. E., Dufour-L'Arrivée, C., Lafond, J. A., Lalancette, R., & Brodeur, J. (2008). Carbon dioxide emissions by urban turfgrass areas. *Canadian Journal of Soil Science*, 88(4), 529-532. <http://doi.org/10.4141/cjss07043>
- Piao, S., Liu, Z., Wang, T., Peng, S., Ciais, P., Huang, M., . . . Tans, P. P. (2017). Weakening temperature control on the interannual variations of spring carbon uptake across northern lands. *Nature Climate Change*, 7(5), 359-363. <http://doi.org/10.1038/nclimate3277>
- Wang, T., Liu, D., Piao, S., Wang, Y., Wang, X., Guo, H., . . . Zhao, Y. (2018). Emerging negative impact of warming on summer carbon uptake in northern ecosystems. *Nature Communications*, 9(1), 5391. <http://doi.org/10.1038/s41467-018-07813-7>
- Tang, R., He, B., Chen, H. W., Chen, D., Chen, Y., Fu, Y. H., . . . Yang, Y. (2022). Increasing terrestrial ecosystem carbon release in response to autumn cooling and warming. *Nature Climate Change*, 12(4), 380-385. <http://doi.org/10.1038/s41558-022-01304-w>
-

L77-78: I do not quite follow the logic. Even if we assume that CO₂ emissions from soil

respiration are a driver of net increases in CO₂ in the atmosphere (which could be the case if decomposition was stimulated), wouldn't GPP also increase and potentially offset at least some of this? Isn't the goal of irrigation to increase plant production?

The net effect combines respiration and GPP. We agree GPP will increase and offset some of the release. Therefore, this study assessed to how much degree the GPP will offset the release. Our result shows that even with the generally increased GPP happens cover CONUS, the respiration increase outpaced the increase of GPP and lead to the overall additional release. We would like to point out that GPP is the result of photosynthesis *in plants*, while most respiration comes from *soil*. Because they happen at different places, their rates can change differently, especially under human inferences. People irrigate intentionally to increase plant production, but meanwhile, unintentionally increase the CO₂ efflux from soil. It is the exact purpose of this study to address this knowledge gap.

L80: which continent?

The reference literatures [23, 24] (in the initial submission) investigated the irrigation cooling effect over CONUS and 100 global cities. The text has been revised to avoid possible confusions.

L84: NC has broad readership. As such, I suggest defining what WRF stands for and what this model does? Only a narrow subset of readers will be familiar with this.

In the revision, we added a brief introduction to WRF model in the method section to improve the overall readability.

L88: How widely adopted is this approach in practice? I am not familiar with any cities doing this at scale with an intention of mitigating the UHI.

In this sentence, we meant to say that irrigation has been widely studied as a mitigation strategy for cooling; while our focus is a much under-explored field: its impact on urban biogenic CO₂ fluxes. We modified the text in revision.

In fact, urban irrigation has been adopted as a nature-based solution in many cities' climate action plans, especially for the mitigation of extreme heat during heatwaves. Melbourne has implemented such action to cool the city (<https://www.melbournewater.com.au/water-and->

environment/climate-change/urban-cooling). We noticed that the real-world implementation of irrigation is rather limited compared to the extensive research effort. This is primarily caused by the considerations of water conservation, maintenance cost, and lack of awareness on the benefits and drawbacks of irrigation. Our study showed that in addition to the cooling effect, irrigation affects city carbon budget, which needs to be considered in decision-making for a climate-ready smart city (Livesley et al., 2021).

Irrigation is also necessary to support the growth of vegetation. As long as the city promotes the adoption of green infrastructures for adaptation and mitigation purposes, it will automatically implement urban irrigation. As we mentioned in the previous response, people irrigate intentionally to maintain the greenness and cooling, but meanwhile, unintentionally increase CO₂ release from soil. Our study investigated the irrigation-induced intentional *and* unintentional impacts on the urban environment, therefore provided a holistic understanding to the question.

Reference:

Livesley, S. J., Marchionni, V., Cheung, P. K., Daly, E., & Pataki, D. E. (2021). Water smart cities increase irrigation to provide cool refuge in a climate crisis. *Earth's Future*, 9(1), e2020EF001806. <http://doi.org/10.1029/2020EF001806>

L89-92: I agree that there could be important CO₂ implications of increasing the latent heat flux across an urban landscape, however I think the important implication is with reduced energy demand for cooling, not changes in biogenic fluxes. The former is the important component or co-benefit for policymakers to consider. Changes in biogenic fluxes are important for urban CO₂ accounting, but unless I am missing an important detail here changes in biogenic fluxes are not important for a cities net CO₂ emissions (I suspect this is true even if irrigation increases decomposition of soil organic matter).

We agree that the cooling effect saves energy consumption and the carbon emissions embedded in the energy. The assessment can be found in many other studies (e.g., Quaranta et al. 2021; Chen et al. 2023). As we mentioned in the manuscript, the change of biogenic CO₂ flux is severely overlooked. To put our results in context, the additional CO₂ release due to irrigation (0.22 g CO₂ m⁻² d⁻¹) in one summer can offset the carbon reduction by replacing *a quarter*

million gasoline cars with EV cars on road. The detail of this calculation is included in the revision of this manuscript. In this regard, it is extremely critical for city to make good decisions to avoid unintended consequences on the biogenic sector.

References:

- Quaranta, E., Dorati, C., & Pistocchi, A. (2021). Water, energy and climate benefits of urban greening throughout Europe under different climatic scenarios. *Scientific Reports*, 11(1), 12163. <http://doi.org/10.1038/s41598-021-88141-7>
- Chen, M., Jia, W., Du, C., Shi, M., Henebry, G. M., & Wang, K. (2023). Carbon saving potential of urban parks due to heat mitigation in Yangtze River Economic Belt. *Journal of Cleaner Production*, 385, 135713. <http://doi.org/https://doi.org/10.1016/j.jclepro.2022.135713>
-

L96-98: Is there really utility in considering irrigation for arid cities that already experience extreme water limitations and are trying to promote a xeric approach to landscaping?

As answered in the previous comment, if a city promotes the adoption of green infrastructures for adaptation and mitigation purposes, it will automatically implement urban irrigation. Arid cities, such as Phoenix and Los Angeles (LA), have water restrictions; therefore, it is important to use water efficiently and try to avoid any possible unintended consequences. Our modeling results showed irrigation in Phoenix and LA will help cool the city but leads to CO₂ release, which indicates the necessity to optimize irrigation water use and conserve water. The arid cities are generally more sensitive to the additional water from irrigation (Fig. 2e) and are in the environmental tradeoff zones (Fig. 6b). Therefore, there are more opportunities for them to mitigate the effect and alter their status quo from tradeoff to environmental co-benefit (Fig. 2bcd).

Also, if this was truly a continent-scale analysis, why are you focusing the text here on just a few cities from a rather limited distribution of climate zones?

This single sentence gives *example* cities with notable air cooling. Figure 1 also shows the air cooling for all urban area in CONUS (subplot in the middle of Fig. 1). In the manuscript, we also discussed other major cities and showed the results in Fig. 2e and Fig. 6.

L118-120: This statement seems to ignore the fact that aboveground plant tissues are considerable sources of CO₂ from respiration.

In this sentence, we meant to qualitatively describe the different environmental controls of air temperature and soil temperature. In fact, we did not ignore plant respiration in our model. The respiration from plant tissue is considered as the plant internal respiration, which is simultaneously controlled by ambient temperature and soil temperature in the modeling process. However, what the soil temperature affect should be the processes in the soil. therefore, we modified the text to “...*the change of soil temperature mainly influences the respiration processes occur in plant root and soil...*” to avoid possible confusions.

L120-122: This result seems counter to much of what is known about photosynthesis. Sure, the small reduction in temperature might have an impact on the leaf-level biochemical processes that govern photosynthesis, but lower temperatures could also keep leaf temperatures from exceeding optima for photosynthesis AND increased water availability is also likely to increases photosynthesis, particularly when soils are drier and VPD is higher. Why do you think your results produce this outcome?

This specific sentence does not describe our results. Instead, it describes the theoretical qualitative outcome. Therefore, we used term “***expected*** to decrease”. Later, we showed our quantitative simulation results in Figure 4 with all major photosynthesis dynamics considered, including the optimum temperature for photosynthesis mentioned by the reviewer, which was set to 25°C (Ronda et al. 2001 Eq. 8). The other key factors are also considered in the model. For example, photosynthesis will not slow down at high temperature if the soil water is ample (Wahid et al 2007). Figure 4 clearly shows that GPP increased as the net outcome of irrigation, while Figure 2b (black solid line) illustrates the mechanisms. Our results are not contradicting to the existing knowledge or the reviewer’s statements.

We understand that the reviewer perhaps refers to the situation when the plant tissue is damaged due to prolonged drought or extreme heat. In that case, irrigation will relief heat stress and help plants maintain normal growing condition. However, an observation-based study pointed out the even with ample irrigation, greenness reduced significantly during dry years (high VPD), showing a decoupled trend between irrigation and plant activity (Quesnel et al.

2019). This study directly refutes the statement “*increased water availability is also likely to increase photosynthesis, particularly when soils are drier and VPD is higher*”. Here, we argue that this situation only happens occasionally during extreme events. Our simulation spans over three summers and the results are presented as the daily mean, which represents the situation in a typical summer day. Under normal situation, the heat tolerance in plants will likely follow Wahid et al. (2007). Again, plant physiological function is extremely complicated to quantify. It is worth in-depth investigation on the plant’s response under extreme conditions in urban environment, but the discussion on this problem is beyond the scope of this study, where we focused on the compound impacts of irrigation on thermal and carbon environments.

References:

- Ronda, R. J., de Bruin, H. A. R., & Holtslag, A. A. M. (2001). Representation of the canopy conductance in modeling the surface energy budget for low vegetation, *Journal of Applied Meteorology*, 40(8), 1431-1444. [https://doi.org/10.1175/1520-0450\(2001\)040<1431:ROTCCI>2.0.CO;2](https://doi.org/10.1175/1520-0450(2001)040<1431:ROTCCI>2.0.CO;2)
- Wahid, A., Felani, S., Ashraf, M., & Foolad, M. R. (2007). Heat tolerance in plants: An overview. *Environmental and Experimental Botany*, 61(3), 199-233. <https://doi.org/10.1016/j.envexpbot.2007.05.011>
- Quesnel, K. J., Ajami, N., & Marx, A. (2019) Shifting landscapes: decoupled urban irrigation and greenness patterns during severe drought. *Environmental Research Letters*, 14, 064012. <http://doi.org/10.1088/1748-9326/ab20d4>
-

Similarly, I would imagine that in arid landscapes, increasing soil moisture could actually have a stimulatory effect on soil respiration. Thoughts on why your results are different? (I see you get at some of this later on)

“Additional soil moisture stimulates respiration” is one of the key conclusions in our study. Figure 5 clearly showed the related results. Meanwhile, we found that this is not always true for all cities and highlighted the spatial variation as the other key conclusion. We also demonstrated the mechanisms in Figure 2c and 2e. Again, our results are not contradicting the current knowledge, but took one step closer to the underlying dynamics that lead to the outcome.

L135: Negative impact on what?

The text has been modified to improve the clarity.

L141: What is the relative change (i.e., % difference) compared to the no irrigation scenario?

After irrigation, GPP_u and R_u increased 1.4% and 5.1% on averaged over CONUS cities, respectively. We did not show the relative change is because the biogenic carbon pool is very large. Though the relative changes seem to be small, the actual variations are significant comparing to the emission amount from the other sectors. In the revision, we provide a comparison between the irrigation-induced NEE change to the annual carbon reduction by adopting electric cars to show the role of irrigation on the overall city carbon budget.

Also, you express these data with per day as the time variable. What does this equate to over the course of the growing season? Year?

To convert the unit, we can multiply the daily value by the number of days. For example, during summer months (defined as May 1st to Aug 31st, 123 days), the total change of NEE_u is 27.06 g m⁻². Seasonal variation of photosynthesis and respiration is significant. As our simulation only covers warm months, providing annual statistics will not be reasonable.

In your methods you indicate that model bias is 1-2 orders of magnitude higher than these differences. How might that confound interpretation of the data?

The high uncertainty in the estimation and modeling is a key point we underscored in this study. We also addressed that there is no “ground truth” for urban biogenic CO₂ fluxes for the moment. The validation, however, is based on the current availability of the spatial gridded data, which estimates from various approaches and with different underlying assumptions. For example, the latest SMUrF urban dataset (Wu et al. 2021) assumes cloud-free conditions all year around. VPMGPP used MODIS data, which sometimes has substantial errors and biases (Wang et al. 2017). The bias between SMUrF and VPMGPP is also large in urban regions. The model bias we disclosed in this study represents the *difference* between VPMGPP and our simulation (reference case). Therefore, we cannot conclude that our results are “accurate or not” based on the products with their own uncertainties. It is noteworthy that, in the revision, we evaluated our

model performance using root mean squared error (RMSE) in addition to the mean absolute bias (MAB) used previously. The model RMSE is $6.5 \text{ gCO}_2 \text{ m}^{-2} \text{ d}^{-1}$ ($1.7 \text{ } \mu\text{mol m}^{-2} \text{ s}^{-1}$) against VPMGPP, which is comparable to the prior process-based modeling studies on biogenic carbon fluxes at daily (Wu et al., 2021) and annual scale (Madani et al., 2017). The previously reported “as high as $25 \text{ gCO}_2 \text{ m}^{-2} \text{ d}^{-1}$ ” is the largest bias observed in the simulation. Model error in most area is significantly lower than this value. The figure below shows the distribution of model bias as MAB.

The validation also illustrates that our simulation can reflect the dynamics of thermal and carbon environments over a large region, which implies our model is *precise*. The scientific question we would like to investigate is “what are the impacts of urban irrigation”; therefore, the study focused more on the change before and after irrigation. Since the model is precise, the general trends (dT_{2m} from CONUS, and $d\text{NEE}_u/dT$ from 20 cities) will not be largely affected by model accuracy. Once there were reliable data serving as the “ground truth”, the modeling framework could be re-calibrated to be more *accurate*. Unfortunately, to the best of our knowledge, there is no known, well-validated, and comparable dataset to use. There are only few point-scale measurements in some cities, which are already discussed in the manuscript. The large uncertainties showed in this study imply the notable scientific gap in estimating urban biogenic processes.

To improve the manuscript, we include a short discussion in the revision on the limitation of this study per the reviewer’s suggestion.

Reference:

Wu, D., Lin, J. C., Duarte, H. F., Yadav, V., Parazoo, N. C., Oda, T., & Kort, E. A. (2021). A model for urban biogenic CO₂ fluxes: Solar-Induced Fluorescence for Modeling Urban biogenic Fluxes (SMUrF v1). *Geoscientific Model Development*, 14(6), 3633-3661.

<http://doi.org/10.5194/gmd-14-3633-2021>

Wang, L., Zhu, H., Lin, A., Zou, L., Qin, W., & Du, Q. (2017). Evaluation of the Latest MODIS GPP Products across Multiple Biomes Using Global Eddy Covariance Flux Data. *Remote Sensing*, 9(5), 418. <https://doi.org/10.3390/rs9050418>

Madani, N., Kimball, J. S., & Running, S. W. (2017). Improving global gross primary productivity estimates by computing optimum light use efficiencies using flux tower data. *Journal of Geophysical Research: Biogeosciences*, 122(11), 2939-2951.

<http://doi.org/https://doi.org/10.1002/2017JG004142>

L226: This does not make sense to me. A key goal of irrigating urban systems should be cooling the city and mitigating the UHI. Afterall, this is what will improve public health and reduce electricity demand for cooling. This recommendation seems flawed to me and if embraced as policy would be counterproductive.

We thank the reviewer's feedback. Possible confusions may exist in this specific sentence. We meant to discuss the influence of irrigation on GPP_u; therefore the "excessive air cooling" refers to the partial effect of temperature on GPP_u (i.e., lower temperature will reduce GPP_u). This specific sentence alone does not serve as a policy recommendation either. In the revision, we modified the text to improve the clarity and avoid possible confusions.

L233-234: I'm not sure anyone actually irrigates when it the soil is wet, that would be a waste of money and water resources.

We totally agree with the second half of this statement. However, over irrigation is a common issue reported in numerous literatures, such as Los Angeles (Livak et al. 2017), Phoenix (Kindler et al. 2021), north Utah (Shurtz et al. 2022), to name a few. Interestingly, these cities are all in semi-arid/arid region and usually under water constraints. Though it may sound non-practical and to irrigate when the soil is wet, it actually happens. In fact, most automatic irrigation systems in urban gardens are not equipped with soil moisture sensors. The irrigation is

metered or timed regardless of weather and soil conditions. Many cities recommend their residents to adopt smart irrigation systems to prevent over-irrigation. Some takes a step forward and makes the installation of smart irrigation systems as a task in their climate action plans.

Our study showed that over irrigation will not only waste water resources, but also unnecessarily leads to the additional release of CO₂. The conclusion aligned with the current call for the smart irrigation systems and suggested optimizing irrigation can save water and carbon simultaneously. This study can help the city understand the underlying dynamics and achieve the overall environment benefits.

Reference:

Litvak, E., Manago, K. F., Hogue, T. S., and Pataki, D. E. (2017), Evapotranspiration of urban landscapes in Los Angeles, California at the municipal scale. *Water Resource Research*, 53, 4236– 4252, <https://doi.org/10.1002/2016WR020254>

Kindler, M., Vivoni, E. R., Pérez-Ruiz, E. R., & Wang, Z. (2022). Water conservation potential of modified turf grass irrigation in urban parks of Phoenix, Arizona. *Ecohydrology*, 15(3), e2399. <https://doi.org/10.1002/eco.2399>

Shurtz, K.M.; Dicataldo, E.; Sowby, R.B.; Williams, G.P. (2022) Insights into efficient irrigation of urban landscapes: Analysis using remote sensing, parcel data, water use, and tiered rates. *Sustainability*, 14, 1427. <https://doi.org/10.3390/su14031427>

Reviewer #3 (Remarks to the Author):

The authors present a very thorough WRF model simulation of the impact of soil moisture controlled irrigation upon air temperature and net ecosystem exchange of CO₂ in urban green spaces for 12 metropolitan regions of USA. The manuscript is generally well written and clearly the method is innovative and thorough.

We thank the reviewer's positive feedbacks on our study. We revised and improved the manuscript per the reviewers' comments and suggestions. Please find the point-to-point responses in the following text.

However, there is a central concern with how the term 'Trade off' is being communicated in this context. The cooling benefit and CO₂ exchange in these cities are effectively being viewed as being equally important – but they are not. The cooling benefit is considerable, targeted and meaningful, whereas the CO₂ exchange increase is negligible and inconsequential when global radiative forcing is considered and that is the only way to view CO₂ emissions – globally. The authors must place the 0.22 g m⁻²d⁻¹ in context, a global context, because we can all place the 0.26 °C in context, a global or city context. To do this, the sum total increase in CO₂ emissions should be compared to annual anthropogenic emissions in CONUS for examples – not that will give researchers and decision-makers an opportunity to assess whether this is a 'trade-off' to worry about. Don't get me wrong, I think this is a great study. But by spinning this to show that with irrigation you can get an important cooling benefit for very like concomitant increase in CO₂ emission from urban vegetation – now that is an impactful study – not just an academic one.

We agree that it will be more meaningful to put our results in a national or global context. First of all, the irrigation-induced change in CO₂ flux (0.22 g CO₂ m⁻² d⁻¹) is significant in comparison to the latest estimate of the posterior 5-year annual mean global atmospheric CO₂ growth rate of 0.41 gCO₂ m⁻²d⁻¹ (converted from 5.35 PgC yr⁻¹ over global land area to flux) (Jin et al., 2021). In addition, the change in biogenic carbon emission is significant when comparing to the ongoing carbon reduction effort from the anthropogenic sector. It is therefore more reasonable to compare the change to change. In the revision, we did a conservative estimation showing that the 0.22 g m⁻²d⁻¹ additional release from urban green spaces can offset

the effort of replacing **0.25 million** gasoline cars with battery electric vehicles (BEV). Comparing to the massive effort in adopting EV from city to federal level, the significance of urban green spaces is clearly overlooked. Most city officials, policy makers, and even researchers have not realized the significance of urban green spaces in carbon budget. We revised our manuscript per the reviewer's suggestion.

Details of the estimation (also see the method section in revised manuscript):

Assuming 15% of the total urban areas in US (University of Michigan, 2022) implement irrigation as modeled. The $0.22 \text{ gCO}_2 \text{ m}^{-2}\text{d}^{-1}$ additional release accumulates to 8.9×10^3 tons CO_2 per day ($0.22 \text{ gCO}_2 \text{ m}^{-2}\text{d}^{-1} \times 15\% \times 2.7 \times 10^{11} \text{ m}^2 \times 10^{-6} \text{ ton/g}$) in a conservative estimation. To put this number into context, we compared it to the carbon reduction effort from transportation electrification. According to the national average data from DOE research centers and labs (DOE 2009, 2016, 2021), replacing a gasoline vehicle by a battery electric vehicle (BEV) can save 4.4 tons CO_2 per year or 12.2 kg CO_2 per day due to the reduced on-road emission. In this case, the $0.22 \text{ gCO}_2 \text{ m}^{-2}\text{d}^{-1}$ release offset the effort of replacing **0.25 million gasoline cars to BEVs**. This does not even account the carbon emissions embedded in EV manufacturing and charging infrastructures. Note that the annual BEV sales (full year) in US are 0.6 million and 0.8 million units in 2021 and 2022.

Reference:

Center for Sustainable Systems, University of Michigan. (2022), "U.S. Cities Factsheet."

Retrieved from: <https://css.umich.edu/publications/factsheets/built-environment/us-cities-factsheet>

Jin, Z., Tian, X., Han, R., Fu, Y., Li, X., Mao, H., & Chen, C. (2021). A global CO_2 flux dataset (2015–2019) inferred from OCO-2 retrievals using the Tan-Tracker inversion system. *Earth System Science Data Discussion*, 2021, 1-40. <https://doi.org/10.5194/essd-2021-210>

U.S. Department of Energy - Argonne National Laboratory. (2009), Well-to-wheels energy use and greenhouse gas emissions analysis of plug-in hybrid electric vehicles. Retrieved from: <https://publications.anl.gov/anlpubs/2009/03/63740.pdf>

U.S. Department of Energy - National Renewable Energy Laboratory. (2016), Emissions associated with electric vehicle charging: Impact of electricity generation mix, charging Infrastructure availability, and vehicle type. Retrieved from:

https://afdc.energy.gov/files/u/publication/ev_emissions_impact.pdf

U.S. Department of Energy - Alternative Fuels Data Center. (2021) Emissions from electric vehicles. Retrieved from: https://afdc.energy.gov/vehicles/electric_emissions.html

Another more minor issue is that there has been no attempt to consider the CO₂ exchange associated with management of green spaces, even though an initial premise of the study was to increase GHG emissions associated with management of building spaces for cooling.

Please find below comments made as I read through the manuscript.

From a broader perspective, the irrigation-induced environmental change will eventually affect the management of building spaces for cooling and the associated carbon emissions. Our study focused more on the biogenic sector. In the revision, we modified the abstract and introduction sections to better focus on the biogenic exchange from urban vegetations and the impacts of irrigation.

Abstract

L24 The abstract does not clearly define the mechanism of positive climate-carbon feedback in the abstract. I assume (have to) that the authors refer to higher temperatures leading to greater building air conditioning use? and therefore greater GHG emissions associated with energy production to run A/Cs, or plant and soil respiration, or both/all?

We meant to say the variation of carbon exchange from biogenic sector (urban green spaces) is large enough to either contribute or counteract the net carbon emissions to the atmosphere depending on the management. We modified the abstract to better articulate this point in the revision.

L26 Please provide in the abstract an indication of how (method) you investigate the impact of green space irrigation on temperatures and CO₂ exchange". Are you modelling, measuring or meta-analysing?

In the revision, we modified the abstract to show that we quantify the impact via modeling approach.

L28 what is this “irrigation-induced environmental co-benefit” of which you write? Is it perhaps related to cooling – if so, then say that as ‘environment co-benefit’ could be many things.

The irrigation-induced environmental co-benefit refers to the two major outcomes led by irrigation, which are 1) heat mitigation; and 2) more CO₂ capture from irrigated green spaces (Li and Wang, 2021). We also use this definition throughout the manuscript as well as the abstract by saying “...sustainable environmental strategies can mitigate both heat and carbon emissions.” In the revision, we addressed the environmental co-benefit of irrigation in abstract, introduction and result sections for better clarity.

L31 Respiration of what? soil, humans, plants? Remember the reader has not read the paper yet, the abstract must be intelligible on its own.

We thank the reviewer to point this out. In the revision, we revised the abstract thoroughly to make it intelligible on its own.

Revised abstract:

Global climate changes, especially the rise of global mean temperature due to the increased carbon dioxide (CO₂) concentration, can in turn, result in higher anthropogenic and biogenic greenhouse gas emissions. This potentially leads to a positive loop of climate-carbon feedback in the Earth’s climate system, which calls for sustainable environmental strategies that can mitigate both heat and carbon emissions, such as urban greening. In this study, we investigate the impact of urban irrigation over green spaces on ambient temperatures and CO₂ exchange across major cities in the contiguous United States. Our modeling results indicate that the carbon release from urban ecosystem respiration is reduced by evaporative cooling in humid climate, but promoted in arid/semi-arid regions due to increased soil moisture. The irrigation-induced environmental co-benefit in heat and carbon mitigation is, in general, positively correlated with urban greening fraction and has the potential to counteract climate-carbon feedback in the built environment.

L40 delete 'the'

L42 changes in global climate rather than global climate changes?

We fixed these typos in revision.

L50 the way this sentence is currently worded it is not strictly true - AnCO₂ does not represent the largest carbon flux to the atmosphere – it does, however, represent the greatest carbon flux that drives the increase in radiative forcing in the atmosphere.

We agree with the reviewer. Soil respiration is a larger terrestrial carbon source than anthropogenic emissions. We meant to say among the potential drivers to climate change, AnCO₂ is arguably the largest carbon flux. The text has been rephased to improve the clarity.

L47 to 56 – this is a logic breakdown – fossil fuels are not burned in buildings to run air conditioners – they are burned in power stations, away from urban areas, to create electricity that is used to run air conditioners in urban areas.

There is a possible confusion in the original text. We meant to say “...more fossil fuel consumption for building cooling...”. In the revision, we modified these texts to better define and “carbon-climate feedback” and avoid possible confusions.

L58 I would strongly disagree that nature-based solutions have been extensively adopted to counteract environmental impacts concomitant with global urbanization. Yes, they have been studied and written about BUT they have not been extensively adopted, especially not urban greening, as many western towns and cities are going backwards in urban green cover and urban forest cover.

We understand that cities in the arid and semi-arid climate regions may face water restrictions, which prevents them having more green spaces. Whether or not adopting nature-based solutions (NbS) and what kind of NbS to adopt are essentially decision-making questions, which should be based on scientific evidence. Our study investigated the impact of one NbS from dual perspectives in terms of the cooling benefit and the role on carbon capture, which updates the knowledge about NbS. In the revision, we rephased the text to better reflect the status quo of the implementation of NbS in the introduction section and further highlighted the motivation of this study.

L82 – rather limited to a few cities? – Citation 24 studied 100 global cities?

In this sentence, we meant to say the studies on the “irrigation-induced impact on CO₂ flux” are limited, while Citation 24 investigate the cooling potential instead. The text has been revised to improve clarity.

L86 and L91 pick one term. Either thermal-carbon OR heat-carbon. Then use throughout.

The text has been revised to improve clarity.

L95 CONUS? Define the first time you use.

In the revision, we defined “CONUS” at its first appearance.

L95 What are you irrigating in these 12 metropolitan regions? The whole city? All urban green space? Public and private green space? How much are you irrigating? What time of day are you irrigating (at night)? How does WRF simulate irrigation? This may well be in the methods section – but please tell the reader here in a just a few words some of these things – they want to know.

The details of irrigation scheme were described in the method section. We modified the text per the reviewer’s suggestion to include a brief description of irrigation scheme in the main text for better clarity.

Figure 2, panel a) does not indicate higher background CO₂ concentration. Also, higher than what? indicate this.

Urban regions have higher ambient CO₂ concentrations (or CO₂ dome in previous literatures) than their rural counterparts due to the intensive anthropogenic emissions in the built environment. However, the primarily focus of this study is evaluating the impacts from urban irrigation. Since our simulation did not show the impacts from CO₂ concentration level, we decided not to show it in Figure 2a to avoid potential confusions at the beginning.

Figures 4 and 5 – are both of these figures really necessary? Perhaps supplementary – it is the NEE that you can talk to when you get to tradeoffs.

We agree that it is the change of NEE_u showing the net effect and the overall outcomes. But more importantly, we emphasized the different pathways leading to the distinctive outcomes and their spatial variability. Figs. 4 and 5 showed the separate contributions from photosynthesis and respiration, corresponding to the mechanisms showed in Fig. 2bcd. From this perspective, they are the major findings of this study. We discussed these two figures thoroughly in section “*The impact of urban irrigation on CO₂ exchange*”. Therefore, we decided to include these two figures in the main text rather than as supplementary information.

L134 – okay you present the CO₂ exchange impact in g CO₂ m⁻² d⁻¹, which is fine BUT not very informative to the reader. It is very easy for all readers to place the cooling impact of 0.26 °C from urban greenspace irrigation in context with say predicted global warming of 1.5°C. However, what you need to do is place this CO₂ exchange in context with global C exchange (daily or annually).

L182 - The reason being is that this so-called trade-off, I suspect, is not really a trade-off to worry about at all. The cooling benefit is considerable, targeted, and meaningful, whereas the CO₂ exchange increase is negligible and inconsequential when global radiative forcing is considered and that is the only way to view CO₂ emissions – globally. If I am wrong, and even if I am not, please, place this irrigation impact on CO₂ exchange in context for the reader. This appears to be a very detailed academic modelling exercise – but please indicate the real impact of these CO₂ flux in a global C flux context.

We combined these two comments that questioned the quantity of the CO₂ exchange presented in this study and answered as the follows:

We thank the reviewer for this constructive suggestion. In one previous response, we have demonstrated the irrigation-induced change of CO₂ fluxes are significant. We further noted that the global biogenic carbon pool is very large, which also holds true in urban environment where the biomass density is usually considered less than the natural areas. Our study showed that any small perturbations of this large pool will be significant when comparing to the other sectors, such as the carbon reduction from transportation electrification. In the revision, we put our analysis to a broader context and emphasized the large variation of biogenic CO₂ exchange.

Another issue is that there has been no attempt to consider the CO₂ exchange associated with management of green spaces, even though an initial premise of the study was increase GHG emissions associated with management of building spaces for cooling. There are several studies that have indicated the CO₂ sequestration by urban vegetation is offset by the combined impact of management events and embedded energy associated and respiration flux.

We understand that CO₂ sequestration by urban vegetation may be offset by the CO₂ release in the maintenance activities, such as the fossil fuel consumptions when operating machines for lawn mowing, tree branch clipping, foliage blowing, etc. In previous responses, we noted that from a broader perspective, the irrigation-induced environmental change will eventually affect the management of building spaces for cooling and the associated carbon emissions. But our study has a narrower focus on the biochemical activities from the biogenic sector, while the management is treated as an external forcing. In the revision, we modified the text in introduction section to better scope our focus of this study.

L223 unintended carbon release is so small in comparison to the associated cooling benefit – this is at the heart of the misleading nature of this study.

In the comparison to the EV adoption case, we showed that the unintended carbon release is not trivial at all. We noticed consensus on the significant role of urban green spaces in carbon budgeting and the impact of anthropogenic forcings on the green spaces has not been reached, which further demonstrates the importance of our study. We revised the manuscript to provide a solid context for our argument.

L226 ‘avoid excessive air cooling’? really?

Possible confusions may exist in the original manuscript. Our analysis and previous literatures showed that the degree of air-cooling increase with the increase of soil moisture. That is to say, profound air cooling is usually associated with excessive soil water (and over irrigation). In this sentence (and the sentence after), we meant to say urban irrigation should be controlled to avoid over irrigation for water saving and further to optimize the benefit of cooling without causing unintended CO₂ release. The discussion is limited to the mechanisms that affect the biogenic CO₂ exchange rather than “the cooling of the city”. The text has been revised to improve clarity.

L245 – 256 I completely agree – it is very complex and models are inadequate to simulate these process interactions yet, let alone simple irrigation-temperature-carbon interactions. As such, this study must not better represent and communicate the impact of irrigation of CO₂ exchange in a global radiative forcing context, otherwise this study is just preventing us from getting to good urban water and green space decision making sooner.

We again agree with the reviewer that the results should be discussed under a commonly known context. But we underscore the significance of biogenic carbon exchange on the overall carbon budgeting. Please see our revised manuscript with an explicit comparison to the EV adoption case, which provides a solid context for our results. We also argue that reducing anthropogenic emission alone cannot be the solution to climate change. We should embrace all the possibilities and investigate every potential measure. Though nature-based solutions have become more practical, and a part of the climate action plans in many cities, most city officials, policy makers, and even researchers have not realized its significance and underlying impacts on carbon budgeting. In this regard, the contribution of our study is meaningful. It provides a modeling approach to quantify the urban biogenic CO₂ exchange, reveals mechanisms of the irrigation controls on NEE, demonstrates the significance role of urban green spaces, and leads to many relevant future works. In the revision, we modified the text to better describe the contributions on science and decision-making, as well as the caveats and future works based on this study.

REVIEWER COMMENTS

Reviewer #2 (Remarks to the Author):

Overall, I appreciate the revisions to this paper and L301-33, in particular. This is an interesting study that clearly represents a massive effort to take on a difficult, but important scientific problem. While I think the content of the paper is worthy of publication, I still have some concerns that were also raised by other reviewers. I firmly agree with the sentiments of reviewer 3 that this study, although quite interesting, is not properly comparing CO₂ emissions from increased respiration with the cooling effects of increased evapotranspiration. Furthermore, I do not think the revised version adequately addresses this concern which was raised by two reviewers. This paper can have important policy implications and I fear that as written it conveys a message that irrigation exacerbates the climate change problem. As a well-mixed GHG, CO₂ acts at global scales while temperature acts at local scales. It would be helpful to compare the cooling effects in W/m² locally (i.e., the city-scale) with the W/m² increase in radiative forcing of increased CO₂ emissions from respiration distributed across the planet. Furthermore, as the authors note, there is a dearth of ground validation data to which these models can be compared (or even parameterized). As with all models, these results need to be taken with a healthy grain of salt and particularly so in this case because of the lack of validation data or even a well-established understanding of these ecophysiological processes in urban areas. Furthermore, unless I am missing an important point, from a mass balance perspective, increases in CO₂ losses from respiration cannot persist in perpetuity without increases in GPP and allocation of C to soils. However, removing an ICE vehicle from the road would represent a “permanent” loss of an emission source. Given this, is it fair to compare C fluxes from these two sources without including caveats? My main suggestion for making this paper publishable and to minimize the likelihood of media articles and policies being based on misinformed “take-home messages” would be for the authors need to better and more clearly reconcile the C emissions from respiration v cooling problem I describe in this paragraph and in comments below.

L63: I think this sentiment comes across as being too certain and should be toned down...perhaps with “may lead to...” with a nod to the role of changes in GPP (e.g., longer, warmer growing seasons, increased CO₂, etc.)

L144-146: I still have a hard time with this result. Work in temperate forests in rural and urban environments indicate that tree CO₂ uptake can be a lot lower during times of heat stress and several studies point to optimum temperatures for photosynthesis between 20 and 28°C in the temperate region. I could be wrong, but I think some process-based models like PNET use ~24°C as an optimum temperature for photosynthesis. I realize that the authors provide a paper for ag crops that might suggest something different, and perhaps they are right. But at a minimum this highlights differences among plant functional types and also uncertainty about the temperature-photosynthesis relationships. Other work has shown that growth of temperate trees in rural and urban forests declines as levels of heat stress increase (see Reinmann et al. below). If those findings are widespread, wouldn't that suggest that the cooling benefits of irrigation could actually help to increase rates of C uptake by trees in cities? Regardless, I think there needs to be discussion about uncertainties in the modeled photosynthesis

results of this paper. I am not necessarily suggesting further uncertainty analyses in a formal sense, but rather text that highlights the different possible responses of vegetation C uptake to reducing UHI.

Cunningham SC and Read J. 2002. Comparison of temperate and tropical rainforest tree species: Photosynthetic response to growth temperature. *Oecologia* 133(2): 112-119

Mau A et al. 2018. Temperate and tropical forest canopies are already functioning beyond their thermal thresholds for photosynthesis. *Forests* 9, 47

Scafaro et al. 2023. Rubisco deactivation and chloroplast electron transport rates co-limit photosynthesis above optimal leaf temperature in terrestrial plants. *Nature Communications* 14:2820.

Reinmann et al. 2020. Urbanization and fragmentation mediate temperate forest carbon cycle response to climate. *Environmental Research Letters* 15: 114036.

L156-161: Why is Boston responding the opposite way as all of the other mesic cities? This is true even compared to nearby cities like NYC where PFTs, tree species composition, and management tend to be quite similar. Also, the authors indicate that this is validated by field data on lines 182-183, however the results described in these studies (at least in Decina et al.) a) should be relevant for NYC and other mesic cities and b) did not explicitly quantify the role of irrigation.

L229-230: Is that because of differences in soil texture or is soil texture not a factor in the model?

L250-252: See references above. Below certain temperature thresholds cooler temps could certainly reduce C uptake, but I suspect that many summer days in most of these cities are near, at, or above optimum temperatures for photosynthesis, no?

L264-268: But if you apply 0.22 to total urban land area (~3% of land area) and 0.41 to total land area, doesn't that make the increase in CO₂ from respiration in cities rather trivial?

L272-275: This number, while large, is infinitesimal compared to the nearly 300 million cars in the U.S. It seems that reduced energy demand from cooler cities would have a bigger impact on C emissions, no? I think that if this argument is to stay in the paper it needs to be paired with an estimate of reduced fossil fuel emissions needed for cooling with reduced UHI. I suspect, but could be wrong, that in that context the climate change benefits of irrigation will be larger than the "costs."

L302-316: I appreciate this section.

Reviewer #3 (Remarks to the Author):

The authors have done a thorough job of responding to reviewer comments and suggestions, drawing upon a wealth of literature to substantiate their statements.

One final suite of comments from myself would be that yes, a short title is a good title, but please can the authors consider a more exacting and appropriate title such as "Can Urban Irrigation Help Counteract Carbon-Climate Feedback in Cities?"

Help is important because urban irrigation can only contribute to this. Cities is important as we are talking about the Urban energy and carbon balance only.

Similarly, in the abstract, please add "help" to the final sentence.

We sincerely thank the editor and reviewers for their constructive feedback and help in improving the quality of this manuscript. Below are the point-to-point responses to all reviewers' comments. All changes and clarifications are incorporated and highlighted in the revised manuscript.

Responses to the comments from Reviewer #2:

Overall, I appreciate the revisions to this paper and L301-33, in particular. This is an interesting study that clearly represents a massive effort to take on a difficult, but important scientific problem. While I think the content of the paper is worthy of publication, I still have some concerns that were also raised by other reviewers.

We thank the reviewer's additional comments and constructive feedback and help in improving the quality of this manuscript. In this round of revision, we modified and rephased certain paragraphs to answer the concerns from reviewers. Below please find our point-to-point response to the reviewer's comments.

I firmly agree with the sentiments of reviewer 3 that this study, although quite interesting, is not properly comparing CO₂ emissions from increased respiration with the cooling effects of increased evapotranspiration. Furthermore, I do not think the revised version adequately addresses this concern which was raised by two reviewers.

From our understanding, in the last round revision of the manuscript, reviewer #3 requested to put our estimation into a global context, because people are aware of a reference of change in temperature (~2°C) but lack of sufficient consensus on the amount of carbon exchange. We agreed with this argument and provided comparisons with the adoption of BEV and the CO₂ growth rate. The other two reviewers were satisfied with the last round of revision.

Meanwhile, we would like to point out that our analysis indicates that the changes in temperature and carbon exchange are strongly entangled processes. Therefore, it may not be technical feasible to directly and quantitatively compare which option is more important and effective to combat climate change, for instance, (a) to reduce average temperature by 0.2 °C via evaporative cooling, vs. (b) to reduce carbon emission by 0.2 gCO₂ m⁻²d⁻¹, as they are qualitatively and dimensionally different. We do not think it is rational to compare CO₂ flux with water flux, either. In fact, our study implies that one can achieve both simultaneously via the control of irrigation in some areas. More importantly, we illustrated in other areas, to pursue cooling from irrigation will induce increased CO₂ efflux and explained the mechanisms.

This paper can have important policy implications and I fear that as written it conveys a message that irrigation exacerbates the climate change problem.

Although many existing impactful studies at larger spatial scales do imply “irrigation exacerbates the climate change problem” to a certain degree, a more accurate and responsible summary of these studies is “inefficient or improper irrigation may exacerbate the climate change problem”. Our conclusions are aligned with these studies but with a specific and in-depth focus on urban areas, where irrigation can be metered, timed, and managed in a more precise

way. The key findings of our study are the intricate mechanisms of biogenic carbon balance under the impact of irrigation. It further indicates that via proper irrigation scheme, one can achieve environmental co-benefit or at least minimize unintended consequences. Therefore, we recommend optimizing irrigation schemes by conducting localized and high-resolution studies over individual cities.

It is noteworthy that the irrigation scheme in our numerical experiment may not fully reflect the real-world situation, as there is no sufficient literature and documentation for us to fine tune the irrigation scheme over all US cities. The irrigation experiment revealed the mechanistic pathways leading to various outcomes. Some achieved environmental co-benefit, some did not. The overall adverse effect in the US happened to be the average level among the urbanized regions under the representative irrigation scheme we investigated in the study. We neither suggested nor intend to imply whether irrigation is good or bad, instead, we underscored the mechanistic pathways, the variances in amount, and their spatial distributions.

As a well-mixed GHG, CO₂ acts at global scales while temperature acts at local scales. It would be helpful to compare the cooling effects in W/m² locally (i.e., the city-scale) with the W/m² increase in radiative forcing of increased CO₂ emissions from respiration distributed across the planet.

Thanks for this insightful suggestion. Comparing the cooling effects with the radiative forcing of increased CO₂ emissions globally would indeed be very meaningful. However, as noted by the reviewer, it is already challenging to find reliable sources to fully validate the results from US cities, let alone resolving the variances due to the distinctive climate, PFTs, soil textures, management practices, etc., over different continents. A comprehensive comparison will certainly require additional investigations over global cities, which is one of the future directions of our study. We plan to apply the modeling framework to global cities and incorporate relevant processes at the global scale, and preferably with a more realistic design of irrigation scenarios based on our communications with local experts.

We do hope, however, with the attentions raised by these urban focused studies, benchmarking observations and experiments could follow to create synergies and address the uncertainties in the estimation of biogenic CO₂ exchange under anthropogenic and environmental perturbations. We also anticipate incorporating with potential future efforts to explore the monumental science question raised by the reviewer.

Furthermore, as the authors note, there is a dearth of ground validation data to which these models can be compared (or even parameterized). As with all models, these results need to be taken with a healthy grain of salt and particularly so in this case because of the lack of validation data or even a well-established understanding of these ecophysiological processes in urban areas.

In this round of revision, we further discussed the limitations of this study regarding the validation in urban environment. Meanwhile, we underscored the unveiling of the intricate mechanistic pathways and development of the modeling framework as the key findings of our

study, as these are actually contributing to “the understanding of ecophysiological processes in urban areas”. Our revision reads:

“More specifically, the lack of observation-based datasets on urban ecosystem services imposes great challenges to estimate long-term urban biogenic CO₂ exchange at a large spatial scale and hinders the validation of such numerical models.”

“Based on the key findings of this study and its current caveat, we advocate the urban research community for more effort on urban observation and data synthesis in biogenic sectors.”

Furthermore, unless I am missing an important point, from a mass balance perspective, increases in CO₂ losses from respiration cannot persist in perpetuity without increases in GPP and allocation of C to soils. However, removing an ICE vehicle from the road would represent a “permanent” loss of an emission source. Given this, is it fair to compare C fluxes from these two sources without including caveats?

We view the carbon balance on a global scale as an equilibrium in the Earth system. The perturbations, e.g., irrigation, can alter the equilibria, leading to changes in gaseous carbon in the atmosphere. The crucial fact is that the current climate is not in equilibrium. What matters to humanity is the time it will take to reach a new equilibrium, a question that remains unanswered. It can be decades, centuries, or even geological time scales. Indeed, processes like respiration and global warming will not increase indefinitely because of the natural self-regulating mechanisms. The reason why humans are still taking proactive measures rather than waiting for it to be self-regulated, is to try to mitigate the adverse effects of climate change as soon as possible and to prevent irreversible consequences. This consensus has been established among the scientific community and governments globally.

We would like to clarify that EV adoption is not simply “removing an ICE vehicle from the road”. It is to *replace an ICE vehicle with an EV*. Therefore, it is by no means “a permanent loss of carbon emission”, rather, it is a process to replace one big carbon source with a smaller one. Additionally, in the comparison to EV adoption scenario, we explicitly wrote “... *calculation may overestimate carbon savings from BEV...*” as we did not count the life cycle carbon emissions from EV manufacturing, end-of-life (recycle), and the construction of new charging infrastructures. We selected transport electrification because it is arguably the most popular approach for carbon neutral. Compared to the tremendous resources allocated to transport electrification, efforts on urban greenery, landscaping, and managed biogenic processes are too trivial. Therefore, attention is needed for science communities and policymakers about the potential large impacts of these factors.

My main suggestion for making this paper publishable and to minimize the likelihood of media articles and policies being based on misinformed “take-home messages” would be for the authors need to better and more clearly reconcile the C emissions from respiration v cooling problem I describe in this paragraph and in comments below.

We totally agree with the reviewer that precautions are needed to minimize the likelihood of misinterpretation from media. Nevertheless, as responsible researchers, we exercise more control on the quality of our study than over how media articles will interpret the scientific findings in specific context. One useful approach, from our experience, is to closely engage with the media

and collaborate with policy makers and help them walk through the content and understand the conclusions thoroughly. In addition, as stated above, we do not think any content of our study and manuscript is “*misinformed*” from a scientific perspective. Findings of our study clearly revealed the critical role of urban irrigation in altering the carbon balance, offering a versatile modeling approach to evaluate the impact under various conditions. These altogether contribute to the multifaceted approach to combat climate change.

L63: I think this sentiment comes across as being too certain and should be toned down...perhaps with “may lead to...” with a nod to the role of changed in GPP (e.g., longer, warmer growing seasons, increased CO₂, etc.)

We toned down the statement per the reviewer’s suggestion, The text “... *leading to a positive loop...*” is modified to “... *potentially leading to a positive loop...*”.

L144-146: I still have a hard time with this result. Work in temperate forests in rural and urban environments indicate that tree CO₂ uptake can be a lot lower during times of heat stress and several studies point to optimum temperatures for photosynthesis between 20 and 28C in the temperate region. I could be wrong, but I think some process-based models like PNET use ~24 C as an optimum temperature for photosynthesis. I realize that the authors provide a paper for ag crops that might suggest something different, and perhaps they are right. But at a minimum this highlights differences among plant functional types and also uncertainty about the temperature-photosynthesis relationships. Other work has shown that growth of temperate trees in rural and urban forests declines as levels of heat stress increase (see Reinmann et al. below). If those findings are widespread, wouldn't that suggest that the cooling benefits of irrigation could actually help to increase rates of C uptake by trees in cities? Regardless, I think there needs to be discussion about uncertainties in the modeled photosynthesis results of this paper. I am not necessarily suggesting further uncertainty analyses in a formal sense, but rather text that highlights the different possible responses of vegetation C uptake to reducing UHI.

References:

Cunningham SC and Read J. 2002. Comparison of temperate and tropical rainforest tree species: Photosynthetic response to growth temperature. Oecologia 133(2): 112-119

Mau A et al. 2018. Temperate and tropical forest canopies are already functioning beyond their thermal thresholds for photosynthesis. Forests 9, 47

Scafaro et al. 2023. Rubisco deactivation and chloroplast electron transport rates co-limit photosynthesis above optimal leaf temperature in terrestrial plants. Nature Communications 14:2820.

Reinmann et al. 2020. Urbanization and fragmentation mediate temperate forest carbon cycle response to climate. Environmental Research Letters 15: 114036.

We agree that there are uncertainties in modeling of photosynthesis. In the revision, we rephased the sentences to further improve clarity by emphasizing the decreases of photosynthesis and respiration led by irrigation is discussed under normal weather conditions without extreme heat

stress. Considering the wide spatial coverage of our model, the current setup may not be able to capture extreme detailed heterogeneity due to the lack of information to fine-tune the model parameters for different cities across the US. This needs to be considered as a caveat of the study. In the last section of the main text, we fleshed out the variances resulting from the different plant types and species across different locations.

Meanwhile, we'd like to point out that in Line 144-146, we are discussing the qualitative change under normal conditions corresponding to the average local climatic conditions. In this situation, lower temperature slows down photosynthesis and decreases carbon uptake. We do understand that in high or extreme heat, cooling from irrigation can reduce heat stress by preventing leaves from “overheating”. In fact, the direction of impact depends on how the actual leaf temperature deviates from its “optimum temperature”. Below we sketched a diagram below showing the qualitative change of GPP (Rate) with change of temperature (T), as shown in **Figure R1**. The green shaded zone indicates the optimum reaction temperature. For native species, the optimum temperature will be the average temperature in the growing season (i.e., normal climate). In high or extreme heat cases, cooling makes the reaction temperature closer to the optima (arrow A); while if the reaction is already under normal/optimum condition, cooling will push it away (arrow B). In a latter comment, we see the reviewer is aware of this qualitative trend.

Figure R1. Sketch of the nonlinear relationship between temperature and CO₂ uptake rate

L156-161: Why is Boston responding the opposite way as all of the other mesic cities? This is true even compared to nearby cities like NYC where PFTs, tree species composition, and management tend to be quite similar. Also, the authors indicate that this is validated by field data on lines 182-183, however the results described in these studies (at least in Decina et al.) a) should be relevant for NYC and other mesic cities and b) did not explicitly quantify the role of irrigation.

In the subsequent paragraphs of the same section (Line 186-204), we explained the intricate mechanism leading to either increase or decrease of urban respiration. Here, we provide a detailed description on why it is possible that Boston and NYC showed different results. Though the following section is rather qualitative, it would help to understand the quantitative results produced by a suite of non-linear functions in the respiration model. The qualitative understanding is also meaningful considering there is no quantitative experiment available for the moment.

From our modeling results, NYC and Boston have similar changes in GPP (Fig. 4). This is supported by the fact that they have similar PFTs and tree species composition. The difference in dNEE_u is primarily caused by the change in urban respiration (Fig. 5). The change of urban

respiration (dR_u) consists of $\partial R_u / \partial SWC$ and $\partial R_u / \partial T_{soil}$, representing the change of R_u induced by the change of soil water content, and the change of R_u by the change of soil temperature, respectively. In Figure 6b (also Fig. S1), we can see dT_{soil} for Boston (9) and NYC (10) are similar, but the average soil temperature in Boston is ~ 2 -degree cooler than NYC (hollow circle 9 and 10 in Fig. 2e). The soil temperature in both cities is below 20°C , which is also far from the optimum temperature for soil respiration. In this temperature range, the qualitative relationship between T_{soil} and R_u is super-linear (see the sketch in **Figure R2**). Because T_{soil} in NYC (dark red dot) is greater than Boston (dark blue dot), with similar amount change of T_{soil} (three units in temperature axis in **Fig. R2**) and the super-linear relationship, the corresponding $\partial R_u / \partial T_{soil}$ is greater in NYC than in Boston (red bar vs blue bar).

Figure R2. A qualitative sketch of the partial dependency of respiration rate on soil temperature when temperature is lower than the optimum temperature range.

On the $\partial R_u / \partial SWC$ side, SWC in NYC is higher than Boston. SWC values in both cities are relatively high (>0.25) and near saturation. In this range, the partial dependency of R_u on SWC is sub-linear. Therefore, NYC will be less sensitive to the increase in SWC, leading to a smaller value of $\partial R_u / \partial SWC$ than Boston.

Collectively, NYC has a larger value of $\partial R_u / \partial T_{soil}$ (lead to large decrease of R_u), and a smaller value of $\partial R_u / \partial SWC$ (lead to minor increase of R_u) compared to Boston, which has a large increase of R_u due to the increase of SWC but a small decrease in R_u due to soil cooling. In the box of Figure 2c (and only in the box of Fig. 2c), the pathway of NYC is marked by dark solid lines, ending with a blue circle and a downward arrow (shows overall decrease of R_u). The pathway of Boston is marked by the red dashed lines, ending with a red circle and an upward arrow (showing overall increase of R_u).

These qualitative discussions show the fundamental mechanisms that cause the distinctions between two near-by places. Though they might have similar PFTs, tree species composition, and management, the small variance in climate can make a difference. In fact, one of the implications of our study is advocating city-wide observations or experiments to accurately measure the role of urban landscaping practices.

L229-230: Is that because of differences in soil texture or is soil texture not a factor in the model?

Soil texture is parameterized in WRF-UBC with different compositions of sand, clay, and silt in a spatially distributed way. The impact of irrigation on cooling and CO₂ exchange, however, is estimated via all parameters and processes considered in the model. It will not be accurate to simply state that it is the difference in soil texture that leads to this outcome. Yet, soil texture played a role in the estimation.

L250-252: See references above. Below certain temperature thresholds cooler temps could certainly reduce C uptake, but I suspect that many summer days in most of these cities are near, at, or above optimum temperatures for photosynthesis, no?

We agree that the high temperatures of many summer days are beyond the optimum temperature for photosynthesis. But it is worth noting that photosynthesis process is jointly governed by temperature, radiation, and other factors. The peak of radiation (PAR) is out-of-phase with the peak of air temperature both diurnally (by 2-3 hours) and seasonally (by 30-45 days), such that the optimum photosynthesis can still occur during summer days. The process can be further complicated by the fact that many urban trees can adapt to warmer ambient temperatures such that photosynthesis can be enhanced under warming conditions. Thus, the impact of cooling can well follow arrow B, as sketched in Figure R1, so as to reduce the carbon uptake. The model used in this study takes into account of these potentially nonlinear interactions among diverse meteorological variables (temperature, PAR, soil moisture, etc.). We added clarification in the revised text, which reads:

“For a process-based model, it is of vital importance to balance the model complexity and feasibility, sometimes with the sacrifice of neglecting secondary processes such as plant fertilization or excluding the variances led by different plant species. The current simulation may not be able to fully capture these detailed heterogeneity and processes due to the lack of information to fine-tune the model parameters for different cities across the US. This will inevitably lead to uncertainties of the modeling results.”

L264-268: But if you apply 0.22 to total urban land area (~3% of land area) and 0.41 to total land area, doesn't that make the increase in CO₂ from respiration in cities rather trivial?

We provided the numbers of the biogenic carbon in cities versus the total land is to illustrate the magnitude of fluxes (carbon amount per unit area per time) is comparable for urban vs. non-urban land. We would also like to address that this 0.22 gCO₂ m⁻² d⁻¹ release is led by anthropogenic interventions, which can be prevented via strategic irrigation at local scale. It is also worth noting that the unintended release has pronounced spatial variations. Some cities would release more CO₂ due to improper irrigation than others. To further support our argument, in the later analysis of the same paragraph, we contrasted this number with the on-going effort to reduce carbon emissions and illustrated how difficult it is to remove carbon emissions. Compared to EV adoption, the control of urban irrigation is considered as a distributed and nature-based solution that can be easily and practically executed.

L272-275: This number, while large, is infinitesimal compared to the nearly 300 million cars in the U.S.

We would like to point out that the comparison is between *the change* induced by irrigation and *the change* induced by EV adoption. Comparing “*the change*” to “*the total*” is not valid to prove anything. If diving deeper into some additional facts, it is easy to find that of the nearly 300 million cars in the US (287 million to be accurate), only around 1% is EV. In 2023, new cars sale is around 15 million, of which 8.3% is EV, taking up 0.43% of the total on-road cars in the US. This includes all types of EVs. Some types may be less “green” than the BEV that we used in the comparison. To some extent, the carbon reduction from EV adoption is not significant either. But it is a consensus that mitigating the impact of climate change needs a multifaceted approach, which can include transport electrification, carbon geo-sequestration, carbon direct capture, etc. Our study explicitly presents the pathway that can enhance the carbon sink power from urban green spaces using irrigation, diversifying the options toward carbon neutrality.

It seems that reduced energy demand from cooler cities would have a bigger impact on C emissions, no? I think that if this argument is to stay I the paper in needs to be paired with an estimate of reduced fossil fuel emissions needed for cooling with reduced UHI. I suspect, but could be wrong, that in that context the climate change benefits of irrigation will be larger than the “costs.”

We totally agree that the comparison can be even more comprehensive if we further quantified the energy consumption saving due to the cooling from irrigation. However, this quantification deserves yet another in-depth investigation, which is not the focus of this study (biogenic sectors). Note that if this quantification study were conducted as recommended by the reviewer, we still need to quantify the impact of irrigation on biogenic CO₂ exchange. Our study contributes exactly to this topic. In fact, cooling from irrigation is a much more well-explored field. But the rare mentioned the “side-effect” of irrigation on biogenic carbon exchange. To this end, our study provides valuable insights into an underexplored field and contributes to developing a more comprehensive understanding of the entangled processes in urban ecosystems.

In addition, as mentioned previously, we neither suggested nor intend to imply whether irrigation is good or bad, instead, we underscored the mechanistic pathways, the variances in amount, and their spatial distributions. Our result also indicates that irrigation leads to environmental co-benefit in some cities (Fig. 3 and Fig. 6ab). Detailed investigations are anticipated at city scale to find the optimum solutions for multiple objectives as envisioned in a previous study (Li et al. 2022).

References:

Li, P., et al. (2022) Multi-objective optimization of urban environmental system design using machine learning. *Computers, Environment and Urban Systems*. (94)101796.

L302-316: I appreciate this section.

We thank the reviewer again for the meticulous reading and insightful suggestions that help to improve the quality of our article.

Responses to the comments from Reviewer #3:

The authors have done a thorough job of responding to reviewer comments and suggestions, drawing upon a wealth of literature to substantiate their statements.

One final suite of comments from myself would be that yes, a short title is a good title, but please can the authors consider a more exacting and appropriate title such as "Can Urban Irrigation Help Counteract Carbon-Climate Feedback in Cities?"

Help is important because urban irrigation can only contribute to this. Cities is important as we are talking about the Urban energy and carbon balance only.

Similarly, in the abstract, please add "help" to the final sentence.

We thank the reviewer for the positive feedback and encouraging comments.

Per the reviewer's suggestion and requirement of the journal, we modified the title of this study as "*The potential of urban irrigation for counteracting carbon-climate feedback*".

The wording of abstract is also modified accordingly.

REVIEWERS' COMMENTS

Reviewer #3 (Remarks to the Author):

I thank the authors for taking on board the few comments I had remaining on the manuscript. I think this will be a good contribution to a global discussion of the role irrigation can play in cooling and GHG exchange in urban ecosystems.

We sincerely thank the editor and all reviewers for their constructive feedback and help in improving the quality of this manuscript. Below please find our responses to the reviewer's remaining comment.

Responses to the comments from Reviewer #3:

I thank the authors for taking on board the few comments I had remaining on the manuscript. I think this will be a good contribution to a global discussion of the role irrigation can play in cooling and GHG exchange in urban ecosystems.

It has been our great pleasure interacting with the reviewer through the constructive review process. We thank the reviewer again for the time, effort, and help in improving the quality of our work.